# The MYB-like protein MylA contributes to conidiogenesis and conidial germination in *Aspergillus nidulans*
Ye-Eun Son[1], He-Jin Cho[1] & Hee-Soo Park [1,2]✉

Myeloblastosis (MYB)-like proteins are a family of highly conserved transcription factors in animals, plants, and fungi and are involved in the regulation of mRNA expression of genes. In this study, we identified and characterized one MYB-like protein in the model organism *Aspergillus nidulans*. We screened the mRNA levels of genes encoding MYB-like proteins containing two MYB repeats in conidia and found that the mRNA levels of four genes including *flbD*, *cicD*, and two uncharacterized genes, were high in conidia. To investigate the roles of two uncharacterized genes, *AN4618* and *AN10944*, deletion mutants for each gene were generated. Our results revealed that *AN4618* was required for fungal development. Therefore, we further investigated the role of *AN4618*, named as *mylA*, encoding the MYB-like protein containing two MYB repeats. Functional studies revealed that MylA was essential for normal fungal growth and development. Phenotypic and transcriptomic analyses demonstrated that deletion of *mylA* affected stress tolerance, cell wall integrity, and long-term viability in *A. nidulans* conidia. In addition, the germination rate of the *mylA* deletion mutant conidia was decreased compared with that of the wild-type conidia. Overall, this study suggests that MylA is critical for appropriate development, conidial maturation, dormancy, and germination in *A. nidulans*.

Myeloblastosis (MYB) proteins are a family of highly conserved transcription factors in most eukaryotes and are involved in a variety of biological functions. Viral MYB (v-MYB), first discovered as an avian retrovirus, causes myeloid leukemia in chickens and regulates transcription in vertebrates[1,2]. Cellular MYB (c-MYB), a homolog of v-MYB, was identified as the proto-oncogene of v-MYB oncogene in humans[1]. Null or overexpression of c-MYB results in fetal hepatic hematopoiesis failure, colorectal cancer, and breast cancer[3–6]. Another two MYB proteins (A-MYB and B-MYB) are also associated with cell survival and development in humans and animals[2]. Similar to the three MYBs in vertebrates, hundreds of MYB-like proteins play vital roles in cell cycle control, cellular morphogenesis, stress response, and primary and secondary metabolism in plants[7–9]. Fungi have several MYB-like transcription factors; for instance, Bas1 is required for spore germination and purine biosynthesis in *Ashbya gossypii*[10]. Similarly, MYB3 is essential for vegetative growth, spore formation, germination, and plant pathogenicity in *Fusarium graminearum*[11]. Moreover, FlbD regulates proper asexual and sexual development in *Aspergillus nidulans*[12].

MYB-like proteins contain a highly conserved DNA-binding domain, the MYB domain, which consists of approximately 50 amino acids that are folded into three α-helices. Based on the number of imperfect repeats, MYB-

like proteins are divided into four groups: 1R-MYB (one repeat), 2R-MYB (two repeats), 3R-MYB (three repeats), and 4R-MYB (four repeats). 1R-MYB proteins contain a partial or single MYB repeat (R1/R2 or R3) and contribute to morphogenesis and development in plants[13,14]. 2R-MYB proteins contain two MYB motifs (R2 and R3 repeats), and most of these proteins are related to the determination of cell fate and regulation of cellular development, stress response, and primary and secondary metabolism in plants[15]. 3R-MYB proteins contain three MYB motifs, namely, R1, R2, and R3 repeats; these proteins regulate cell cycle in humans, animals, and plants[16–18]. 4R-MYB proteins rarely exist in most organisms and are only found in plants. Although all MYB proteins are 3R-MYB in humans, most MYB-like proteins are 2R-MYB in plants. Recent research has shown that *Magnaporthe oryzae* has no 3R-MYB and 4R-MYB proteins but contains a greater number of 2R-MYB proteins than 1R-MYB proteins. Interestingly, one of the 2R-MYB proteins, *Mo*MYB1, affects conidiogenesis and virulence in rice plants. Another 2R-MYB, *Mo*MYB8, is required for fungal growth and stress tolerance[19]. Nevertheless, there is a lack of a comprehensive study of MYB-like proteins in fungi, excluding in *M. oryzae*.

*A. nidulans* is a saprophytic filamentous fungus that lives ubiquitously in soil, crops, seeds, and water[20]. The asexual spore (conidium) of *A.*

¹School of Food Science and Biotechnology, Kyungpook National University, Daegu 41566, Republic of Korea. ²Department of Integrative Biology, Kyungpook National University, Daegu 41566, Republic of Korea. ✉e-mail: phsoo97@knu.ac.kr

*nidulans* is the primary reproductive particle that maintains long-term viability and tolerates environmental stimuli[21,22]. The aerial conidia break their dormant state under favorable conditions, swell, form germ tubes, and grow vegetatively. Moreover, hyphae undergo asexual development and reproduce conidia. This series of processes is mediated by a variety of regulators, such as those involved in development (heterotrimeric G proteins), cAMP-activated protein kinase A (PKA) pathway, and mitogen-activated protein kinase pathway as well as Flb proteins and central asexual regulators (BrlA, AbaA, and WetA)[23]. In conidia, VosA, VelB, and WetA regulate conidial viability, germination, and maturation[24]. Despite previous studies, there is a myriad of gene regulators that remains undiscovered.

In the present study, we determined 21 genes encoding MYB-like proteins in *A. nidulans* and characterized one *myb* gene that is highly expressed in conidia compared with hyphae. **My**b-**l**ike protein **A** (MylA) is a newly identified transcription factor in *A. nidulans*; this protein contains two tandem MYB domains. We herein investigated the role of MylA using multiple phenotypic analyses and genome-wide analyses to provide insights into the regulatory roles of MylA in *A. nidulans*.

## Results

### Twenty-one MYB-like proteins in *A. nidulans*

To identify MYB-like proteins in *A. nidulans*, we scanned the MYB-like domain using the InterProScan tool[25]. In total, 21 genes containing the SANT/Myb domain (IPR001005) were found in the *A. nidulans* genome (Table 1). We further explored the phylogenetic relationship among *A. flavus*, *A. fumigatus*, *F. graminearum*, and *M. oryzae*, in which MYB proteins have been investigated previously (Fig. 1)[19,26]. In total, 11 MYB-like proteins were conserved in all species, and the remaining 10 existed only in three *Aspergillus* species. The size of the 21 MYB-like proteins identified varied from 188 to 2031 amino acids. Analysis of the total domains of the 21 MYB-like proteins revealed one or two MYB domains in each protein; moreover, various non-MYB domains, including zinc finger and helicase, were detected.

For a comprehensive overview of the MYB-like proteins, we classified the 21 MYB-like proteins based on the number and sequence of MYB repeats. The sequences of MYB motifs were aligned with those of *Arabidopsis thaliana*, investigated previously (Supplementary Fig. 1)[15], and domain structures were predicted using SWISS-MODEL (Supplementary Fig. 2). As shown in Table 1, there were no MYB-like proteins having more than three MYB repeats. A total of 14 proteins had one MYB repeat, including 6 proteins with the R1/R2 motif and 8 proteins with the R3 motif. In contrast, seven proteins had two MYB repeats, i.e., R2 and R3 motifs. The average repeat size was 50 amino acids, and each repeat had three α-helixes.

### Characterization of conidia-specific 2R-MYBs

Previous studies have shown that 2R-MYBs act as transcriptional activators as well as repressors and regulate several important aspects of cell fate and development in plants[27,28]. To explore whether 2R-MYBs determine cell differentiation in conidia, we conducted a transcriptomic analysis of the conidia and hyphae of *A. nidulans* wild-type (WT) strain and identified 5502 differentially expressed genes (DEGs) (Fig. 2a and Supplementary Data 1). Analysis of the DEGs of 2R-MYBs (Fig. 2b) revealed that mRNA levels of four 2R-MYB genes, including *flbD*, *cicD*, and two unknown genes (*AN4818* and *AN10944*), were significantly upregulated in conidia. We designated these two uncharacterized genes *AN4818* and *AN10944* as *mylA* and *mylB*, respectively.

To determine the functions of *mylA* and *mylB*, we constructed deletion strains of *mylA* (Δ*mylA*) and *mylB* (Δ*mylB*) in *A. nidulans*. The Δ*mylB* strain exhibited no difference in fungal growth, whereas the Δ*mylA* strain exhibited severe lack of fungal growth and differentiation (Fig. 2c). Therefore, we further evaluated the functions of MylA in *A. nidulans*. MylA was conserved in other *Aspergillus* species, and all of them had two MYB repeats (Fig. 2d). Moreover, the relative expression level of *mylA* was approximately 4-fold higher in conidia than that during vegetative growth and asexual development (Fig. 2e).

### Functions of MylA in the growth and development of *A. nidulans*

MYB-like proteins are known to be involved in cell growth and development in plants and fungi[8,19,28]. To investigate the functions of MylA in asexual development, we constructed the complementary strain of *mylA* (C′

**Table 1 | Identification of MYB-like protein genes in *A. nidulans***

| Gene ID | Name | Location | Protein length (a.a.) | # MYB domains | Class | Position of MYB domain (5′–3′) | Size of MYB domain (a.a.) |
|---------|------|----------|----------------------|---------------|-------|-------------------------------|---------------------------|
| *AN0153* | | ChrVIII | 860 | 1 | 1R-MYB | 667–726 | 60 |
| *AN0279* | *flbD* | ChrVIII | 314 | 2 | 2R-MYB | 6–56, 61–110 | 51, 50 |
| *AN4472* | | ChrIII | 1477 | 1 | 1R-MYB | 872–929 | 58 |
| *AN4524* | | ChrIII | 332 | 1 | 1R-MYB | 239–282 | 44 |
| *AN4527* | | ChrIII | 188 | 2 | 2R-MYB | 95–140, 141–188 | 46, 48 |
| *AN4618* | | ChrIII | 724 | 2 | 2R-MYB | 314–361, 369–434 | 48, 66 |
| *AN4797* | | ChrIII | 615 | 1 | 1R-MYB | 302–355 | 54 |
| *AN4813* | | ChrIII | 863 | 1 | 1R-MYB | 759–814 | 56 |
| *AN5118* | | ChrIV | 314 | 1 | 1R-MYB | 107–154 | 48 |
| *AN5476* | | ChrIV | 208 | 1 | 1R-MYB | 15–70 | 56 |
| *AN5643* | | ChrIV | 1105 | 1 | 1R-MYB | 865–909 | 45 |
| *AN6446* | *cicD* | ChrI | 341 | 2 | 2R-MYB | 7–58, 69–105 | 57, 44 |
| *AN6575* | | ChrI | 587 | 1 | 1R-MYB | 435–476 | 52 |
| *AN6705* | | ChrI | 681 | 1 | 1R-MYB | 377–425 | 49 |
| *AN7174* | | ChrIV | 305 | 2 | 2R-MYB | 4–58, 68–111 | 55, 44 |
| *AN7739* | | ChrIV | 290 | 1 | 1R-MYB | 222–266 | 45 |
| *AN8076* | | ChrII | 2031 | 1 | 1R-MYB | 998–1044 | 47 |
| *AN8377* | | ChrIV | 372 | 2 | 2R-MYB | 16–64, 76–119 | 50, 44 |
| *AN10763* | *adaB* | ChrI | 517 | 1 | 1R-MYB | 81–128 | 48 |
| *AN10944* | | ChrIV | 791 | 2 | 2R-MYB | 9–52, 61–106 | 44 46 |
| *AN12280* | | ChrIII | 406 | 1 | 1R-MYB | 351–403 | 53 |

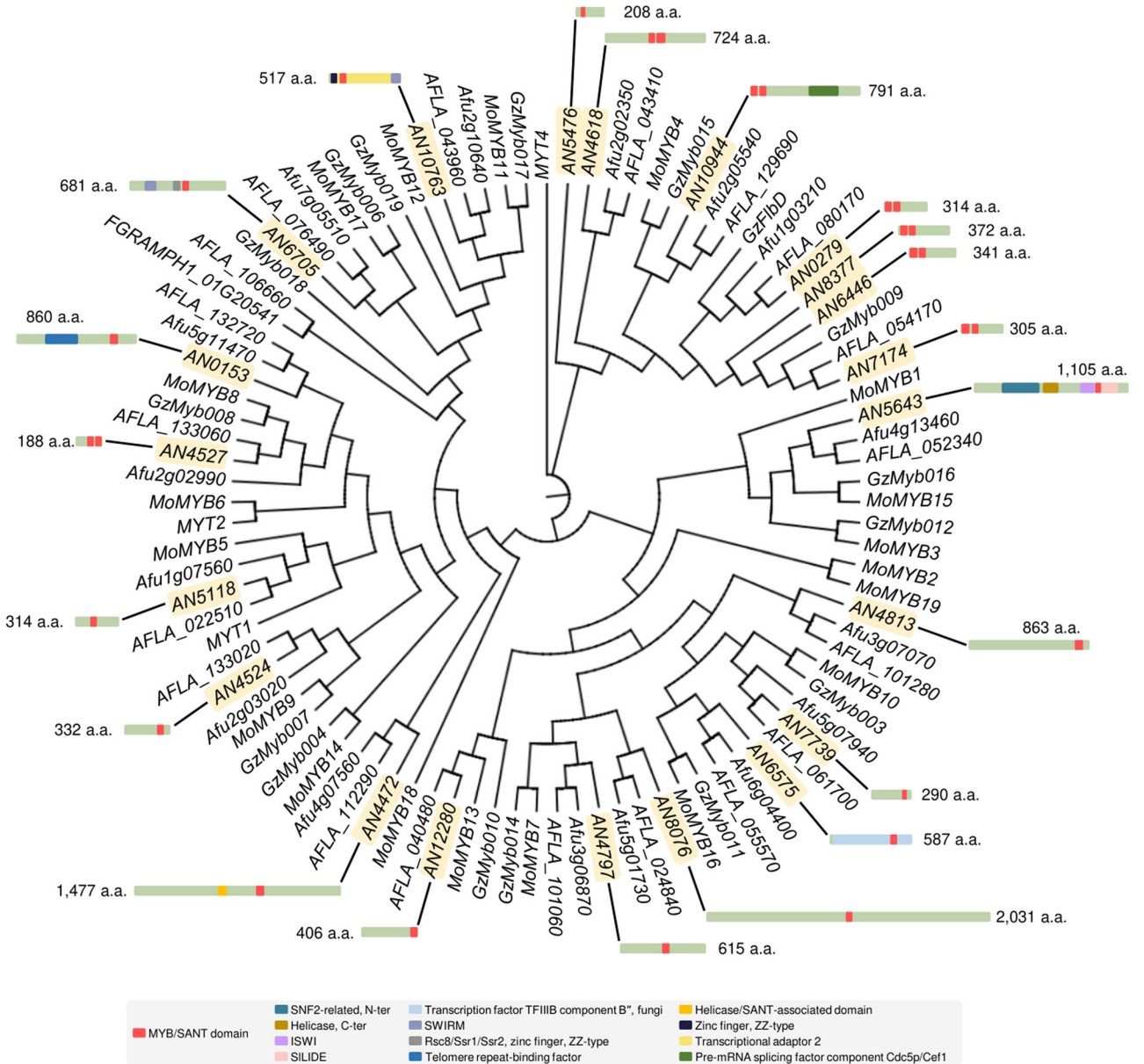

**Fig. 1 | Phylogenetic tree analysis of putative 21 MYB-like proteins.** A phylogenetic tree of MYB-like proteins existed in *Aspergillus nidulans* FGSC4 (AN), *A. fumigatus* Af293 (Afu), *A. flavus* NRRL3357 (AFLA), *Magnaporthe oryzae* 70-15 (Mo), and *Fusarium graminearum* PH-1 (Gz). All sequences were downloaded from FungiDB and aligned using Clustal W and MEGA X software, and a tree was generated in CLC Sequence Viewer 8 using the neighbor-joining algorithm based on the Jukes–Cantor model. The bootstrap for this tree was repeated 1000 times. The architectures of MYB-like domains and other domains in each MYB-like protein from *A. nidulans* are depicted.

*mylA*) and then point-inoculated the WT, Δ*mylA*, and C′ *mylA* strains into solid minimal media containing 1% glucose (MM) for culture at 37 °C for 5 days (Fig. 3a). The Δ*mylA* strain exhibited severely decreased fungal growth, and the fungal colony of the Δ*mylA* strain appeared light green in color. The conidiophores of the Δ*mylA* strain were smaller than those of the WT and C′ *mylA* strains. Quantitative analyses of fungal growth and conidiation revealed that the Δ*mylA* strain exhibited considerably defective fungal growth (Fig. 3b) and decreased number of conidia under both dark and light conditions (Fig. 3c).

To investigate whether *mylA* deletion affects the mRNA expression of central development regulators, we examined the mRNA levels of *brlA*, *abaA*, and *wetA* after induction of asexual development in the Δ*mylA* mutant strain. As presented in Fig. 3d, *brlA* and *abaA* mRNA levels were significantly higher in the Δ*mylA* strain at 24 h after asexual developmental induction than in the WT and C′ *mylA* strains. The *mylA* deletion strain

exhibited higher *wetA* expression than the WT and C′ *mylA* strains at 48 h after induction. These results indicated that MylA is needed for the proper expression of asexual developmental regulators.

To evaluate the roles of MylA in sexual development, we inoculated the WT, Δ*mylA*, and C′ *mylA* strains into solid sexual media (SM) and cultured them at 37 °C for 7 days (Fig. 3e). The Δ*mylA* strain exhibited decreased colony diameter and produced small cleistothecia compared with the WT and C′ *mylA* strains (Fig. 3f, g). Overall, these findings indicated that MylA plays important roles in fungal growth and development in *A. nidulans*.

**Transcriptomic analysis of Δ*mylA* in conidia**
As mentioned above, the mRNA expression of *mylA* was particularly higher in conidia than in the hyphae of *A. nidulans* (Fig. 2e). To further understand the function of MylA in conidia, we performed high-throughput RNA sequencing using the conidia of the WT and Δ*mylA* strains. In total, 4147

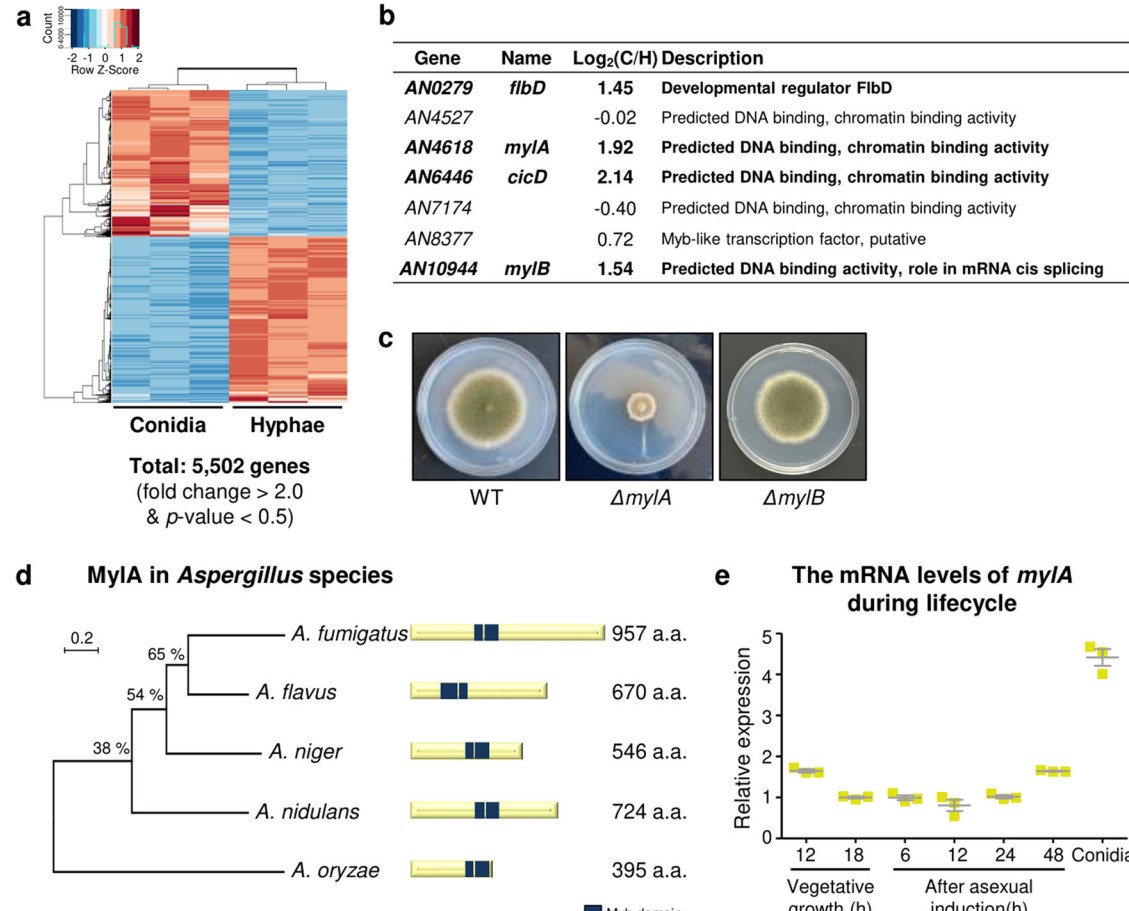

**Fig. 2 | Identification of conidia-specific 2 R MYB-like proteins in *A. nidulans*.** **a** Heat map showing differentially expressed genes (DEGs) in the conidia and hyphae of wild-type *A. nidulans*. **b** Summary of genes encoding the 2 R MYB-like proteins in *A. nidulans*. **c** Fungal colony phenotypes of WT, Δ*mylA*, and Δ*mylB* strains. **d** Phylogenetic tree and domain architecture of MylA homologs in five *Aspergillus* species. A phylogenetic tree of MYB-like proteins existed in *A. nidulans* FGSC4, *A. fumigatus* Af293, *A. flavus* NRRL3357, *A. niger* NRRL3, and *A. oryzae* RIB40. All sequences were downloaded from FungiDB and aligned using ClustalW and MEGA X software, and a tree was generated in MEGA X using the maximum likelihood method based on the JTT matrix-based model. The bootstrap for this tree was repeated 1000 times. The numbers above the branches indicate the percent data coverage for internal nodes following the bootstrap test. The branch lengths, which were proportional to the number of nucleotide substitutions per site, were measured using a bar scale (0.2). **e** The transcript levels of *mylA* during vegetative growth, postasexual development induction, and conidia of wild-type *A. nidulans*.

genes (approximately 37.98% of the genome) were found to be significantly differentially expressed, among which 1854 (16.87% of the genome) were downregulated, and 2320 (21.11% of the genome) were upregulated in the Δ*mylA* conidia (Supplementary Data 2). Gene Ontology (GO) analyses were conducted to predict alterations in the biological process by MylA (Fig. 4a). The significantly downregulated DEGs were related to nucleic acid metabolic process, stimulus/stress responses, cellular component biogenesis, cell cycle/mitotic cell cycle, and transcription by RNA polymerase II. The significantly upregulated DEGs were associated with nitrogen biosynthesis, aromatic compound biosynthesis, carbohydrate metabolism, secondary metabolite biosynthesis (austinol and monodictyphenone), and transmembrane transport. Overall, these findings indicated that MylA is involved in the regulation of mRNA expression of genes associated with fungal cell cycle, response to various stresses, and primary and secondary metabolism in conidia.

## Roles of MylA in conidia

In RNA-seq data, deletion of *mylA* affected mRNA expression of genes involved in stress response (Fig. 4a); thus, we hypothesized that MylA controls stress response in *A. nidulans* conidia. To test this, spore stress analyses were conducted using 2-day-grown conidia of each strain. After exposure to thermal stress (55 °C) for 30 min, the Δ*mylA* conidia showed severely decreased survival rate compared with the WT and C′ *mylA* conidia

(Fig. 4b). Moreover, under oxidative stress (0.1 M $H_2O_2$), the survival rate of the WT and C′ *mylA* conidia was approximately 60% and 70%, respectively, whereas the survival rate of the Δ*mylA* conidia was approximately 25% (Fig. 4c).

Trehalose, a nonreducing disaccharide, is a key factor for resistance against various external stresses[29]. To check whether MylA functions in conidial stress resistance by regulating the content of trehalose, we measured the amount of trehalose and found that the trehalose content in the conidia of the Δ*mylA* strain was reduced by half compared with that in the conidia of the WT and C′ *mylA* strains (Fig. 4d). We next analyzed the transcript levels of trehalose biosynthetic genes in transcriptome data. The mRNA levels of most trehalose biosynthetic genes were considerably downregulated in the conidia of the Δ*mylA* strain (Fig. 4e). To verify this, we performed quantitative reverse transcription-PCR (qRT-PCR) and found that the Δ*mylA* conidia exhibited significantly decreased mRNA levels of all trehalose biosynthetic genes compared with the WT and C′ *mylA* conidia (Fig. 4f).

As trehalose was closely associated with spore viability in *Aspergillus* species[29,30], we evaluated asexual spore viability of the WT, Δ*mylA*, and C′ *mylA* conidia using asexual spores grown for 2, 7, or 10 days. As shown in Fig. 4g, the survival rates of the 2-day-grown conidia of the WT, Δ*mylA*, and C′ *mylA* strains were almost 100%. However, the viability of the 7- or 10-day-grown conidia of the Δ*mylA* mutant strain was decreased compared with that of the WT and C′ *mylA* strains. These findings suggest that MylA is

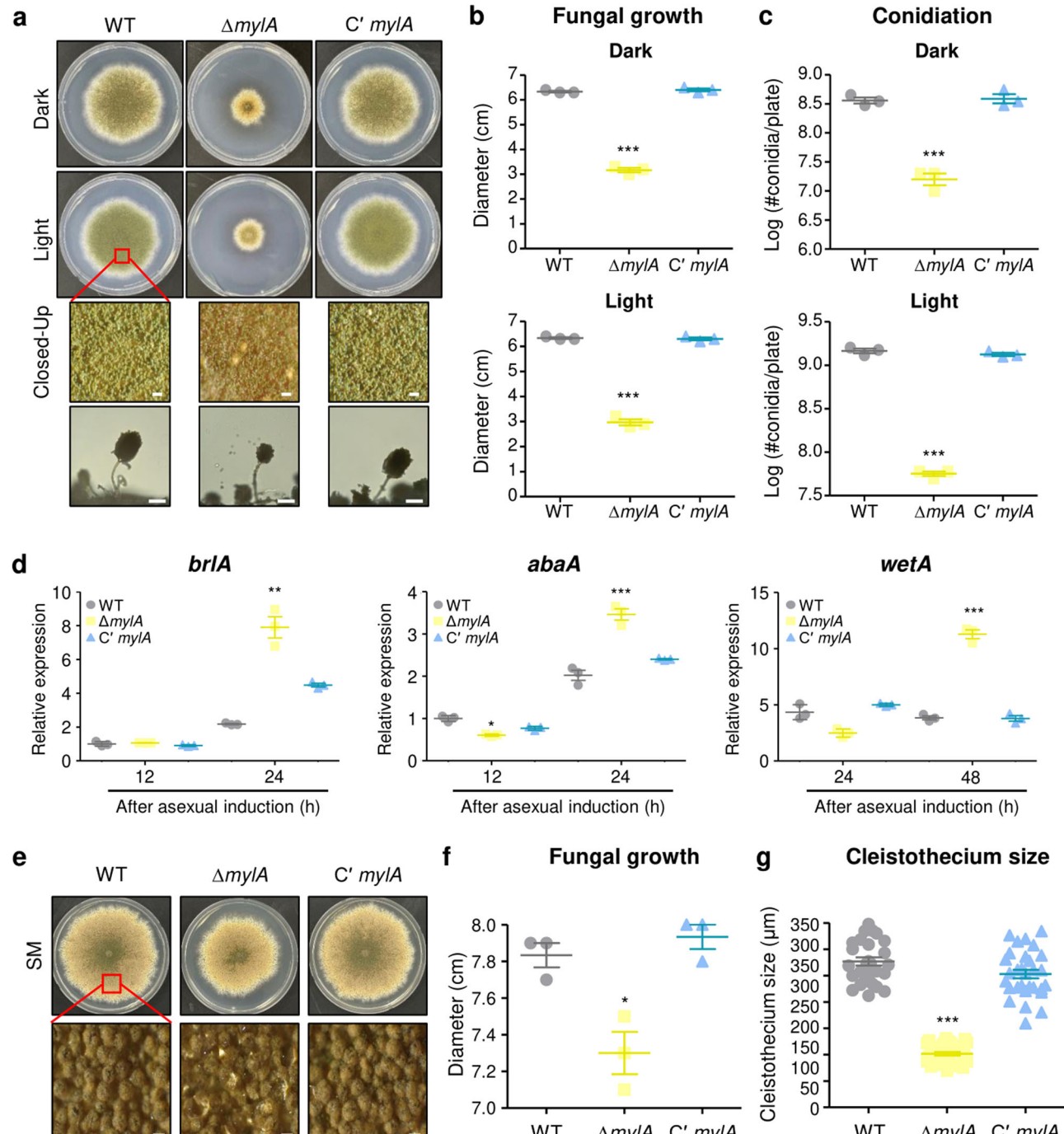

**Fig. 3 | Functions of MylA in fungal growth and development in *A. nidulans*.**
**a** Point phenotypes of WT, Δ*mylA*, and complementary (C′ *mylA*) strains grown on solid minimal media at 37 °C for 5 days under dark or light conditions. Right panels show the magnified images of the middle of the plates and the conidiophore morphology of each strain (bar = 0.25 μm). **b** Fungal colony diameter of WT, Δ*mylA*, and C′ *mylA* strains shown in **a** (***$p < 0.001$). **c** Conidial production of WT, Δ*mylA*, and C′ *mylA* strains shown in **a** (***$p < 0.001$). **d** The expression levels of central development regulators (*brlA*, *abaA*, and *wetA*) were measured after asexual

development induction of WT, Δ*mylA*, and C′ *mylA* strains (***$p < 0.001$, **$p < 0.01$, *$p < 0.05$). **e** Colony phenotypes of WT, Δ*mylA*, and C′ *mylA* strains grown on solid sexual media at 37 °C for 7 days in the dark. Below panels show the enlarged views of the middle of the plates (bar = 0.25 μm). **f** Colony diameter of WT, Δ*mylA*, and C′ *mylA* strains shown in **d** (*$p < 0.05$). **g** Cleistothecium size of WT, Δ*mylA*, and C′ *mylA* strains shown in **d** (***$p < 0.001$). The error bar represents mean ± SEM, and the asterisk indicates significant difference in data (**b**–**d**, **f**, **g**).

essential for conidial trehalose biosynthesis, stress tolerance, and spore viability in *A. nidulans*.

### Effect of MylA on cell wall composition in conidia
The fungal cell wall plays an important role in maintaining the viability of fungal cells and protecting them from various environmental

stresses[31]. To confirm the effect of MylA on conidial cell wall composition, we analyzed mRNA expression of genes related to β-glucan, galactomannan, hydrophobins, and DHN-melanin biosynthesis based on RNA sequencing results. We found that mRNA levels of most genes involved in β-glucan biosynthesis were upregulated in the conidia of the Δ*mylA* strain (Fig. 5a). To verify RNA-seq data, the expression levels of

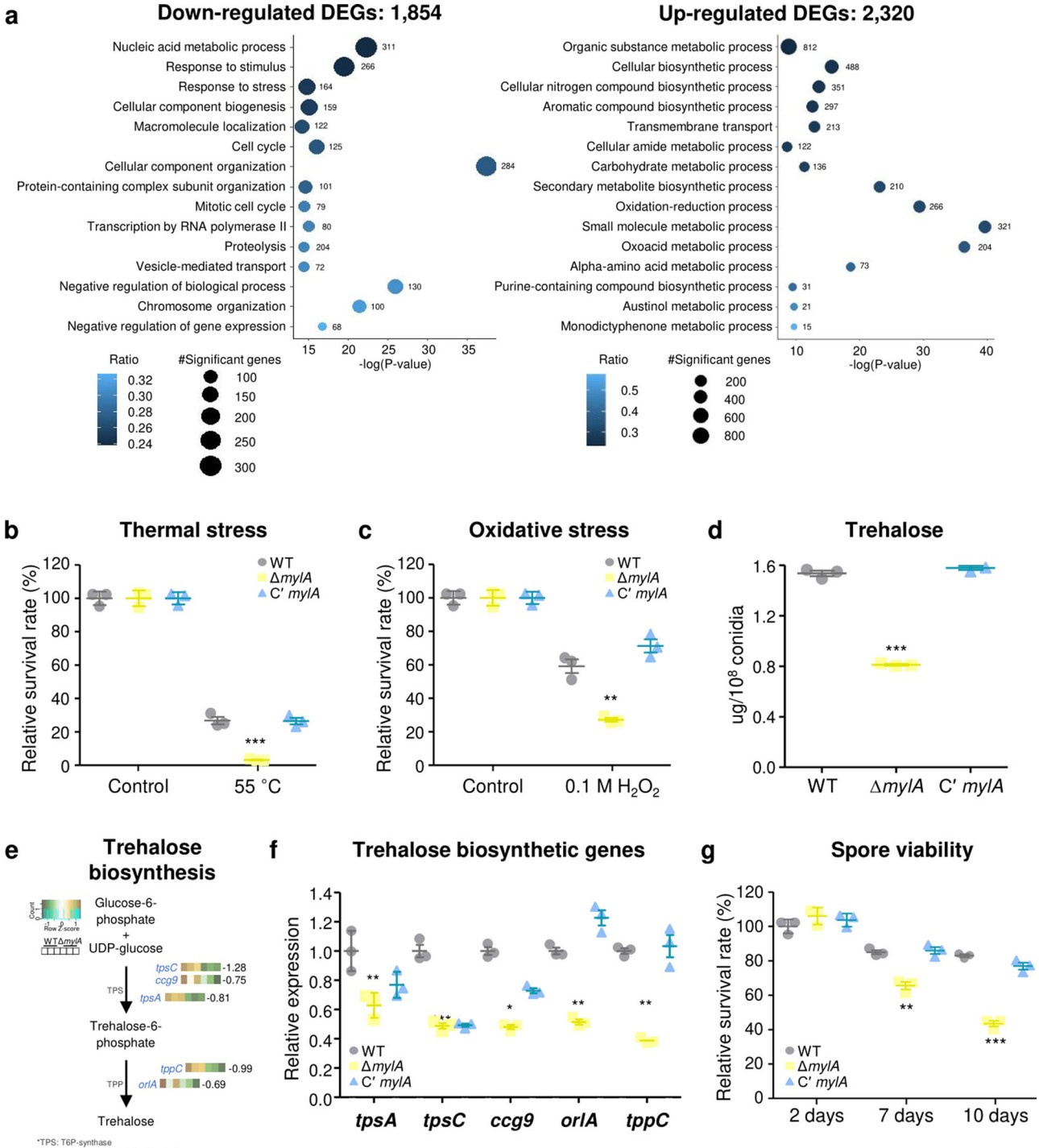

**Fig. 4 | Roles of MylA in asexual spores. a** The results of Gene Ontology analysis affected by MylA. The downregulated (1854) or upregulated (2320) DEGs between WT and Δ*mylA* conidia were analyzed using the Gene Ontology term enrichment method. The gene numbers enriched in each category are indicated to the right of the points. **b** The relative spore survival rate of WT, Δ*mylA*, and C′ *mylA* strains against thermal stress (55 °C for 30 min, ***$p < 0.001$). **c** The relative spore survival rate of WT, Δ*mylA*, and C′ *mylA* strains against oxidative stress (0.1 M H₂O₂ for 30 min, **$p < 0.01$). **d** The amount of trehalose in the conidia of WT, Δ*mylA*, and C′ *mylA*

strains (***$p < 0.001$). **e** Heat map showing the differentially expressed genes involved in trehalose biosynthesis between WT and Δ*mylA* conidia. Log₂ (fold change) of each gene is indicated to the right of the points. **f** Transcript levels of five trehalose biosynthetic genes in the conidia of WT, Δ*mylA*, and C′ *mylA* strains (***$p < 0.001$, **$p < 0.01$, *$p < 0.05$). **g** The relative spore survival rate of WT, Δ*mylA*, and C′ *mylA* strains grown for 2, 7, or 10 days (***$p < 0.001$, **$p < 0.01$). The error bar represents mean ± SEM, and the asterisk indicates a significant difference in data (**b–d, f, g**).

*fksA* encoding β-1,3-glucan synthase were examined via qRT-PCR. We found that the mRNA level of *fksA* in the Δ*mylA* conidia was four times higher than that in the WT and C′ *mylA* conidia (Fig. 5b). We then evaluated β-glucan contents in the conidial surface of each strain. The β-glucan content in the Δ*mylA* conidia was approximately 1.6- and

1.3-fold higher than that in the WT and C′ *mylA* conidia, respectively (Fig. 5c). These data suggest that MylA is required for proper β-glucan biosynthesis in *A. nidulans* conidia. The mRNA expression levels of genes related to galactomannan, hydrophobins, and DHN-melanin biosynthesis were upregulated in *mylA* deletion conidia (Fig. 5d−f).

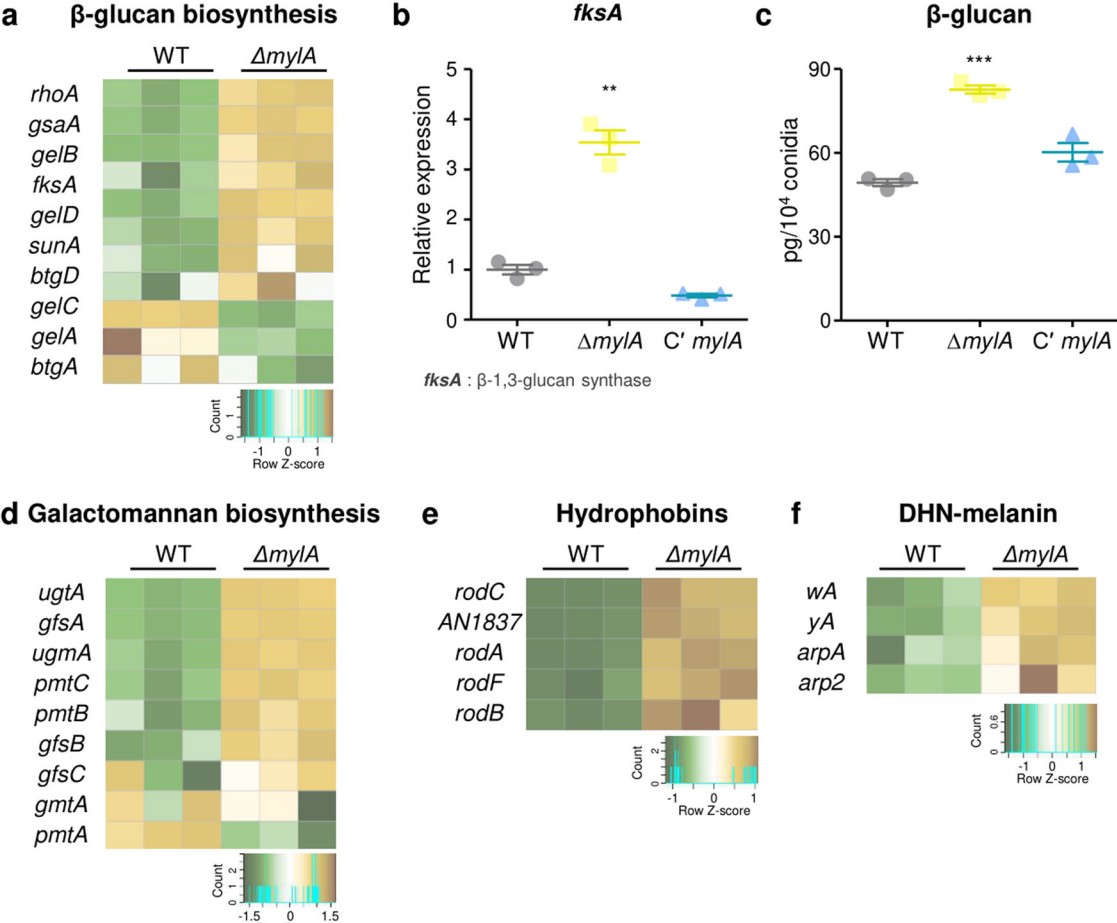

**Fig. 5 | Effect of MylA on cell wall integrity. a** Heat map showing differentially expressed genes related to β-glucan biosynthesis between WT and Δ*mylA* conidia. **b** mRNA expression levels of *fksA* in WT, Δ*mylA*, and C′ *mylA* conidia (**$p < 0.01$). **c** Amount of β-glucan in WT, Δ*mylA*, and C′ *mylA* conidia (***$p < 0.001$). The error bar represents mean ± SEM, and the asterisk indicates a significant difference in data (**b**, **c**). **d**–**f** Heat map showing differentially expressed genes related to galacto-mannan (**d**), hydrophobin (**e**), and DHN-melanin (**f**) biosynthesis between WT and Δ*mylA* conidia.

These results imply that MylA might be involved in regulating conidial wall composition.

### Effect of MylA on conidial germination

The repetitive mitotic cell cycle breaks its dormant state and establishes the axis of polarity, resulting in the germination of *Aspergillus* conidia. When cells meet the size threshold, the ensuing mitosis triggers septation, leading to cell compartmentalization[32,33]. Our transcriptomic analysis indicated that MylA was associated with the mitotic cell cycle (Fig. 4a). To understand the effect of MylA, we investigated the expression levels of 256 genes known to be associated with the mitotic cell cycle. As a result, we identified 95 DEGs between WT and Δ*mylA* conidia. Among them, ~80% (76 DEGs/95 DEGs) were downregulated in the conidia of the Δ*mylA* strain (Fig. 6a). To assess whether the decreased transcript levels of mitotic cell cycle-related genes affected conidial germination, we conducted conidia germination assays. An equal amount of conidia from each strain was spread onto solid MM and incubated at 37 °C. Most WT and C′ *mylA* conidia germinated within 6 h of incubation, whereas the Δ*mylA* conidia started to germinate after 6 h of incubation. Although several Δ*mylA* conidia germinated after 8 h of incubation, their germ tube appeared to be shorter than that of the WT and C′ *mylA* conidia (Fig. 6b). A quantitative analysis of the germination rate by time showed that the conidial germination rate of the Δ*mylA* strain significantly decreased at each time point (Fig. 6c), and most spores of the Δ*mylA* strain completed germination after 8–9 h of incubation. These results implied that MylA affects proper conidial germination by regulating the mitotic cell cycle-associated genes in *A. nidulans* conidia.

### Relationship between MylA and RgsA

RgsA, a regulator of G-protein signaling, suppresses spore germination and vegetative growth by repressing GanB-mediated signaling, which activates vegetative growth through the cAMP/PKA signaling pathway[34,35]. To determine the potential relationship between MylA and RgsA, we generated double-deletion mutants of *mylA* and *rgsA* (Fig. 7a). As illustrated in Fig. 7b, the Δ*mylA* and Δ*mylA* Δ*rgsA* conidia exhibited weak spore germination. Quantitative analysis revealed that the Δ*rgsA* conidia germination rate at 3–5 h was higher than that of the control conidia (Fig. 7c). The germination rate of the Δ*mylA* Δ*rgsA* conidia was similar to that of the Δ*mylA* conidia. These results suggested that the regulation of conidial germination by MylA is unaffected by *rgsA* deletion, and the relationship between MylA- and GanB-mediated signaling should be confirmed through additional studies.

### Discussion

MYB-like proteins are DNA-binding transcription factors that are highly conserved in eukaryotic organisms, including humans, plants, and fungi. These proteins regulate growth, primary and secondary metabolism, and stress response in plants[15]. Genes encoding MYB-like proteins are associated with development, mycotoxin production, and pathogenicity in some fungi[19,26,36]. Although some MYB-like proteins have been functionally characterized, the function of several MYB-like proteins in *Aspergillus* species remains unknown. In the present study, we first analyzed the genome of *A. nidulans* and found 21 genes encoding MYB-like proteins (Fig. 1). Among them, the roles of AdaB (involved in chromatin remodeling), FlbD (upstream developmental activator), and CicD (transcriptional activator of

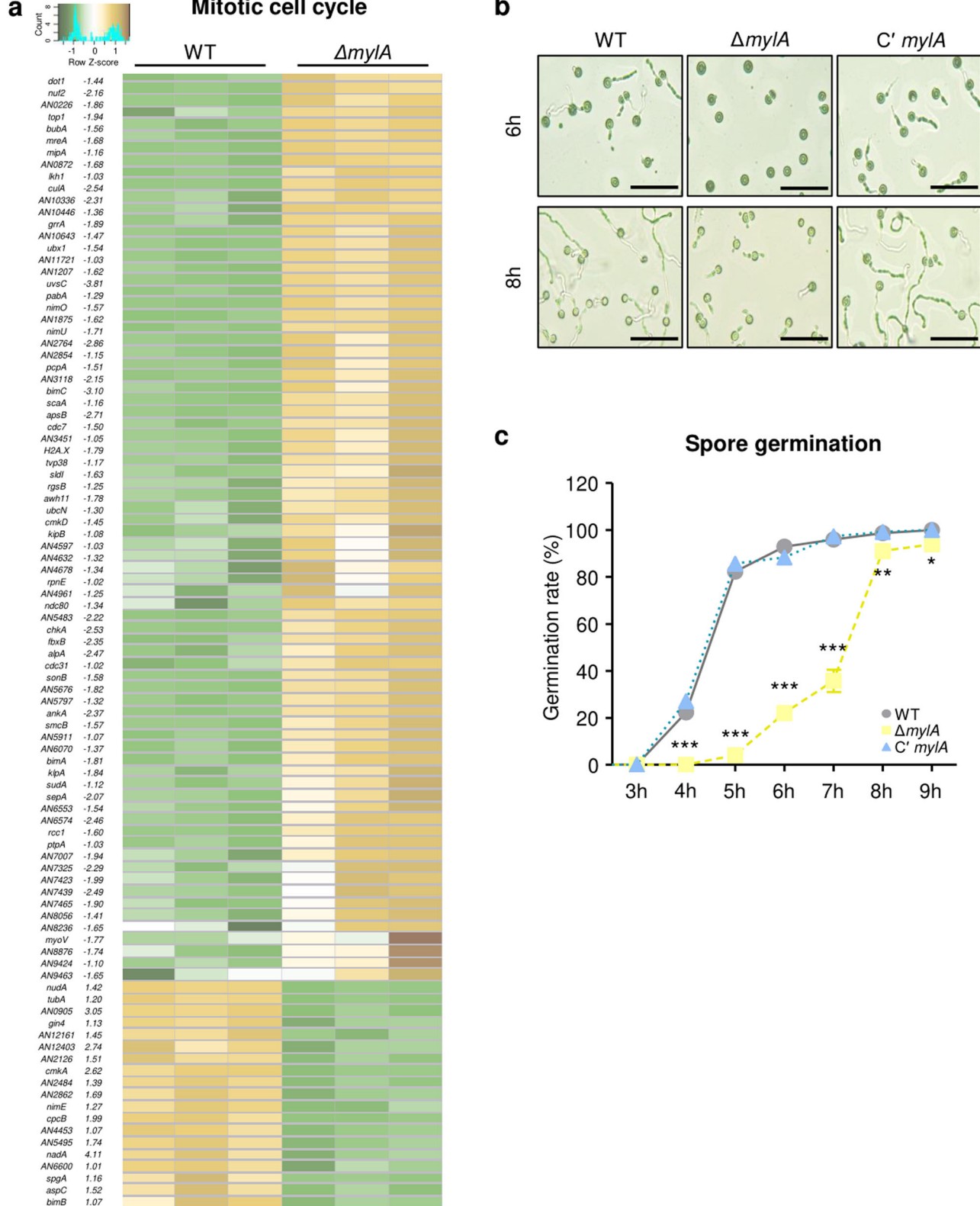

**Fig. 6 | Effect of MylA on mitotic cell cycle and spore germination. a** Heat map showing differentially expressed genes associated with the mitotic cell cycle between WT and $\Delta mylA$ conidia. **b** The microscopic image of conidial germination of WT, $\Delta mylA$, and C′ *mylA* strains after 6 or 8 h of incubation (bar = 50 μm). **c** Graph presenting the germination rate (%) of each strain by time. The asterisk indicates a significant difference between WT and mutant strains (***$p < 0.001$, **$p < 0.01$, *$p < 0.05$).

**Fig. 7 | Double mutant analyses of *mylA* and *rgsA*.**
**a** Fungal colony of control, Δ*mylA*, Δ*rgsA*, and Δ*mylA* Δ*rgsA* strains grown on solid MM at 37 °C for 5 days. **b** The microscopic image of conidial germination of control, Δ*mylA*, Δ*rgsA*, and Δ*mylA* Δ*rgsA* strains after 7 or 9 h of incubation (bar = 50 μm). **c** Graph presenting the germination rate (%) of each strain by time. The asterisk indicates a significant difference between control and mutant strains (***$p < 0.001$, **$p < 0.01$, *$p < 0.05$).

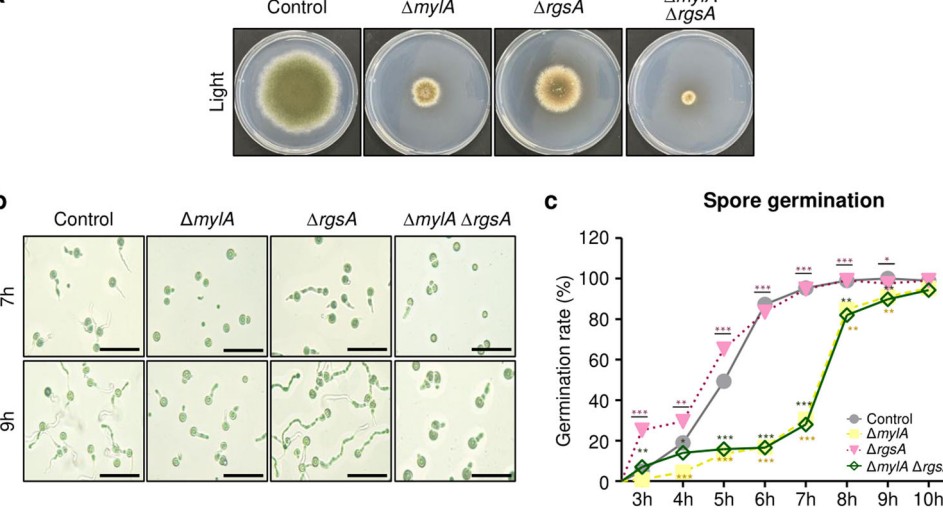

cichorine biosynthesis) were determined[12,37,38] FlbD is involved in asexual differentiation, whereas CicD is involved in cichorine biosynthesis; both these act as antibacterial and anti-HIV agents[12,38]. In the present study, we selected two uncharacterized MYB-like proteins, MylA and MylB, which contained two MYB motifs (R2 and R3 repeats) based on RNA-seq data (Fig. 2a). We found that the absence of *mylA* resulted in severely defective fungal growth and development. However, the Δ*mylB* strain exhibited development that did not significantly differ from that of the WT strain (Fig. 2c). Based on these results, we further explored the roles of MylA in the biology of *A. nidulans*.

In this study, we examined the role of MylA in asexual development. The Δ*mylA* strain exhibited decreased amount of conidia and produced abnormal conidiophores (Fig. 3a). In addition, *mylA* deletion affected the mRNA expression of *brlA*, *abaA*, and *wetA* (Fig. 3d), implying that MylA regulates the transcript levels of genes associated with asexual development. An in-depth mechanistic study is needed to confirm this hypothesis. We found that the mRNA expression of *mylA* was specifically elevated in conidia (Fig. 2e), and thus, we wondered which regulator controls *mylA* mRNA expression in the late stage of asexual development or in conidia. To analyze this, we first investigated the *mylA* promoter sequence and confirmed the presence of one AbaA-response element (5′-CATTCY-3′), one VelB-response element (5′-CCXTGG-3′), and two VosA-response elements (5′-CCXXGG-3′), but not WetA-response element (Supplementary Fig. 3a)[24,39]. We next checked the mRNA level of *mylA* in the deletion mutant strains. We compared the *mylA* transcripts levels with previous RNA-seq data and found no significant difference in the mRNA levels among the Δ*vosA*, Δ*velB*, and Δ*wetA* strains[24]. We then investigated *mylA* transcript in the Δ*abaA* strain and found its increased levels during asexual development (Supplementary Fig. 3b). These findings suggest that AbaA, but not VosA, VelB, and WetA, might control *mylA* expression during asexual development of *A. nidulans*.

Our transcriptome data revealed that MylA positively regulated various genes related to the mitotic cell cycle, even among cell cycles in *A. nidulans* conidia (Fig. 4a). When we further analyzed the mitotic cell cycle-related genes, we found most genes to be downregulated in the conidia of the Δ*mylA* strain (Fig. 7a), among which LAMMER kinase A (LkhA; Log₂FC = −1.03) affects the emergence of germ tube and hyphal polarity by modulating the transcript levels of *nimxcdc2*, which controls the S phase and G2/M DNA damage checkpoint[40]. NimO (**N**ever **i**n **m**itosis; Log₂FC = −1.57) is required to check a cell cycle linking S and M phases and is essential for proper DNA synthesis after progression of the S phase. The *nimO* mutants exhibited delayed germination compared with the WT strain[41]. In addition, the putative NimO-binding partner Cdc7 (Log₂FC = −1.05) has been reported to affect DNA replication[42]. As an

anaphase-promoting complex/cyclosome (APC/C), BimA (**B**locked-**in**-**m**itosis; Log₂FC = −1.81) is essential for spindle poles to promote the onset of anaphase in mitosis. Loss-of-function mutations in *bimA* have been reported to result in growth-arrested cells with condensed chromatin and a short mitotic spindle[43]. BimC, a motor protein (kinesins; Log₂FC = −3.10), plays a role in spindle pole body separation and mitotic spindle formation[44]. SepA (Log₂FC = 2.07) and SepH (Log₂FC = −1.13) enable the formation of cortical actin rings at the surrounding septation and around the tips. In addition, PabA (Log₂FC = −1.29) is required for proper septation during septation and cytokinesis in *A. nidulans*[45]. Overall, these findings suggest that MylA is required for activating almost all stages of mitosis, including the interphase and mitosis.

Previous studies have shown that cell cycle-related genes are involved in spore germination and growth in *Aspergillus* species[33,46]. Dormant conidia form germ tubes and synthesize septum after the first nuclear division. Based on our results, we hypothesized that MylA causes normal spore germination in *A. nidulans*. In fact, the absence of *mylA* led to a lesser rate of conidial germination than that in the WT and C′ *mylA* strains (Fig. 7c). These findings indicate that MylA plays a crucial role in conidial germination by regulating the cell cycle. Nonetheless, further studies are required to investigate the detailed molecular mechanisms of MylA in cell cycle.

MYB-like proteins universally exist in fungi, including *A. nidulans* (Fig. 1), and several MYB-like TFs are highly conserved in fungi and even their functions are conserved across species. For instance, Ada2, a member of the Spt-Ada-Gcn5 acetyltransferase (SAGA) complex, regulates gene expression through histone acetylation and affects stress response in *Saccharomyces cerevisiae*, *Cryptococcus neoformans*, *Candida albicans*, and *A. nidulans*[47–50]. FlbD plays important roles in cell differentiation and asexual/sexual development in *M. oryzae*, *F. graminearum*, and *A. nidulans*[51–53]. Additional analyses of the MylA protein sequence using the BLASTP tool revealed that similar to other MYB-like proteins, MylA was highly conserved in multicellular fungi (Fig. 1) as well as in unicellular fungi. Remarkably, MylA is an ortholog of Liv4 in *C. neoformans*. It has been demonstrated that *Cn*Liv4 is vital for fungal growth, melanization, normal morphology, and lung infection in a mouse model[54]. Although the phyla of *C. neoformans* and *A. nidulans* are different, MylA exhibits similar roles in them, i.e., it is essential for fungal growth and morphology. However, because the cellular roles of MylA homologs remain unclear, further studies are required to provide insights into the role of MylA in the biological processes of other fungi.

To summarize, we investigated the total MYB-like proteins and identified conidia-specific 2 R MYB-like TFs in the model organism *A. nidulans*. Among them, MylA is a newly discovered protein, which is required for fungal growth and development. Based on our phenotypic and

**Fig. 8 | Proposed model for the roles of MylA in _A. nidulans_.** In conidia, MylA plays vital roles in stress resistance, spore viability, and cell wall compositions, thereby affecting spore maturation and dormancy. MylA is also required for proper conidial germination and vegetative growth by coordinating various cell cycle-related genes.

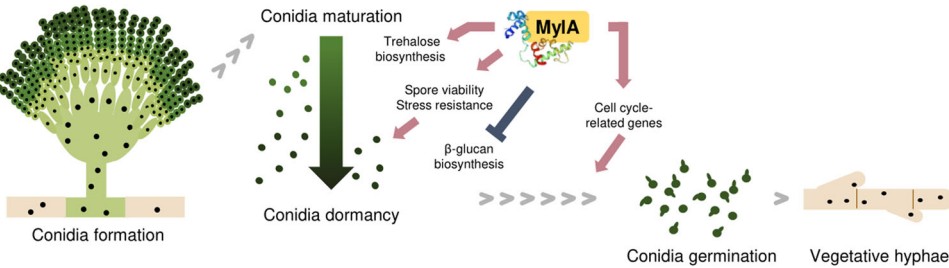

transcriptomic studies, we suggest that MylA is essential for proper conidial maturation and dormancy and positively regulates spore germination and vegetative growth by coordinating various cell cycle-related genes. Thus, we propose a schematic model describing the roles of MylA in _A. nidulans_ (Fig. 8). Despite our new findings, further investigation into the direct regulatory mechanisms of MylA by performing chromatin immunoprecipitation (ChIP) sequencing and combining RNA-seq and ChIP-seq analyses is required to provide novel insights into fungal development and spore properties. In addition, although we focused on conidia-specific 2 R MYB-like TFs, further studies are required to determine the function of other 2 R or 1 R MYB-like proteins in conidiogenesis in _A. nidulans_.

## Methods
### Strains, media, and culture conditions
The _Aspergillus_ strains used in this study are described in Supplementary Table 1. _A. nidulans_ TNJ36 was used as the control strain, and _A. nidulans_ RJMP1.59 was used as the parental strain to construct deletion mutants. For general purposes, _Aspergillus_ strains were cultured in liquid or solid MM[55]. For inducing sexual development, each strain was inoculated onto solid SM and incubated at 37 °C for 7 days[56]. For plasmid constructions, _Escherichia coli_ DH5α was grown in Luria–Bertani medium containing 100 μg/mL of ampicillin (Thermo Scientific, Waltham, MA, USA).

### Identification and phylogenetic analyses of MYB-like proteins
MYB-like proteins in _A. nidulans_ were identified using InterPro 94.0 based on the SANT/Myb domain (IPR001005)[57]. The sequence of each MYB-like protein in _A. nidulans_ was downloaded from FungiDB (https://fungidb.org). The secondary structure of the MYB domain was predicted using SWISS-MODEL[58]. The homologs of each protein were searched using Basic Local Alignment Search Tool in the National Center for Biotechnology Information, including those of _A. flavus_, _A. fumigatus_, _M. oryzae_, and _F. graminearum_. All sequences were aligned using the CLUSTAL W algorithm in MEGA X, and the phylogenetic tree was generated using CLC Sequence Viewer 8 (neighbor-joining method based on the Jukes–Cantor model, bootstrap = 1000).

### Construction of deletion strains and complementary strains
The deletion strain of MYB-like TF genes was constructed using the double-joint PCR (DJ-PCR) method[59], and all oligonucleotides used in this study are listed in Table 2. For homologous fragments, the 5′ and 3′ regions of _mylA, mylB,_ and _rgsA_ were amplified using the primer set 5′ DF:3′ _pyrG_ tail and 5′ _pyrG_ tail:3′ DR, with _A. nidulans_ FGSC4 genomic DNA (gDNA) as the template. The _pyrG_ marker was amplified using the primer set OHS1542:OHS1543, with _A. fumigatus_ Af293 gDNA as the template. All fragments were linked and amplified using the primer set 5′ NF:3′ NR. The resulting PCR products were purified and introduced into the recipient RJMP-1.59 strain (_pyrG89; pyroA4_) according to a PEG-mediated transformation method[55]. At least three independent mutant strains were isolated and confirmed via PCR and restriction enzyme digestion (Supplementary Figs. 4, 5).

The Δ_mylA_ Δ_rgsA_ double-deletion strains were generated via DJ-PCR using the _pyroA_ marker. The 5′ and 3′ regions of _mylA_ were amplified using the primer sets OHS2757:OHS2763 and OHS2764:OHS2760, respectively,

and the _pyroA_ marker was amplified using the primer set OHS1873:OHS1874, with _A. nidulans_ FGSC4 gDNA serving as the template. All fragments were linked and amplified using the primer set OHS2761:OHS2762. The resulting PCR products were purified and introduced into the recipient Δ_mylA_ strain TYE69.1 (_pyrG89_; Δ_mylA::AfupyrG_; _pyroA4_) to produce TYE121.1–3. At least three independent double-deletion strains were confirmed via PCR and restriction enzyme digestion (Supplementary Fig. 5). To analyze the phenotypes among single- and double-deletion strains, the THS30 strain (_pyrG89_; _AfupyrG⁺_; _pyroA⁺_) was used as the control.

Complementary strains for the Δ_mylA_ strain were generated using the _mylA_ cloned plasmid (pYE14.1). For constructing the plasmid, the _mylA_ gene region with its promoter was amplified using the primer pair OHS1675:OHS1676, digested with _Not_I, and cloned into pHS13[60]. The cloned plasmid pYE14.1 was transformed into the recipient Δ_mylA_ strain TYE69.1 to produce TYE72.1–2. The _mylA_-complemented candidates were confirmed via PCR and qPCR analyses (Supplementary Fig. 6).

### Fungal growth and development
Approximately $10^5$ spores of each strain were point-inoculated in solid MM and incubated at 37 °C for 5 days under dark or light conditions. Fungal growth was analyzed based on the diameter of the fungal colony. After harvesting the conidia using 0.02% triton X-100 (Thermo Scientific), their number was counted using a hematocytometer under Leica DM500 (Leica Microsystems, Wetzlar, Germany). For sexual development, each strain was point-inoculated in solid SM and incubated for 7 days at 37 °C in the dark. After washing with 95% ethanol (Merck, Burlington, MA, USA), the size of the sexual structure was measured using a Zeiss Lab.A1 microscope and the AxioVision (Rel. 4.9) digital imaging software (Carl Zeiss, Oberkochen, Germany). All experiments were repeated at least three times.

### Transcriptomic analysis
For the transcriptomic analysis of _A. nidulans_ conidia, 2-day-grown conidia were suspended and filtered using miracloth. In the case of hyphae, $10^5$ spores/mL were inoculated in liquid media and incubated at 37 °C and 200 rpm for 16 h, and the growing mycelia were filtered using miracloth[61].

In brief, all samples were homogenized using a mini-bead beater and TRIzol (GeneAll Biotechnology, Seoul, Republic of Korea), and total RNA was precipitated using isopropanol (Thermo Scientific) and ethanol. The RNA dissolved in DEPC-treated water (Bioneer, Oakland, CA) was treated with RQ1 RNase-free DNase (Promega, Madison, WI, USA) and purified using RNeasy Mini Kit (Qiagen, Hilden, Germany). The RNA samples were sent to Theragen Bio (Suwon, Republic of Korea), and the RNA library from each sample was constructed and sequenced on Illumina Novaseq 6000. The resulting Fast files were trimmed with Trimmomatic, and filtered reads were annotated with the _A. nidulans_ FGSC A4 transcriptome using the aligner STAR v.2.3.0e software. DEGs were evaluated and normalized using the DESeq2 method. DEGs were defined based on an absolute value ($log_2$(fold change) of ≥1 and a _p_ value of <0.05. GO analyses were performed using the R package and FungiDB. The processed data were subjected to Fisher's exact test, and results were plotted using ggplot in R studio (version 4.2.2). All RNA-seq experiments were conducted using at least three biological replicates.

## Table 2 | Oligonucleotides used in this study

| Name | Sequence (5'→3')ᵃ | Purpose |
|---|---|---|
| OHS1542 | CCTGGTCTTTGGTTTGGTACACC | 5' *AfupyrG* marker_F |
| OHS1543 | CGACTGGCAGGAGATGATCC | 3' *AfupyrG* marker_R |
| OHS0197 | TCCCAGCAATCTGCCCTTGC | 5' *AfupyrG_R* |
| OHS1873 | ATCTCATGGGTGCTGTGCG | 5' *AnipyroA* marker_F |
| OHS1874 | TTGCATCGCATAGCATTGCATTG | 3' *AnipyroA* marker_R |
| OHS1875 | CCACGGATGATCGACCTGAC | 5' *AnipyroA_R* |
| OHS1255 | CTGTGGCTGGTGCTTATCG | 5' *AN4618* DF |
| OHS1256 | *GGCTTTGGCCTGTATCATGACTTCA*GACGCACTCCAGAGCTGAA | 3' *AN4618* with *AfupyrG* tail |
| OHS1257 | *TTTGGTGACGACAATACCTCCCGAC*CGGGAGAGTATGGTTGATTAACGA | 5' *AN4618* with *AfupyrG* tail |
| OHS1258 | CATGGAGGCAGCATCTGC | 3' *AN4618* DR |
| OHS1259 | CTGTAGCTCGCTCAACTGAGA | 5' *AN4618* NF |
| OHS1260 | GCCTCGTGGCATGTTTCC | 3' *AN4618* NR |
| OHS1261 | CTGGGACAAGTTGAGAAGCG | 5' *AN4618* RT_F |
| OHS1262 | CACTGCTGCCGTGATCTAAC | 3' *AN4618* RT_R |
| OHS2261 | GCGGCGAAGCTTAGTGTC | 5' *AN10944* DF |
| OHS2262 | *GGCTTTGGCCTGTATCATGACTTCA*GCTGAGGTTCGGCGAAAC | 3' *AN10944* with *AfupyrG* tail |
| OHS2263 | *TTTGGTGACGACAATACCTCCCGAC*TGAGGCGCGCAAGGTATA | 5' *AN10944* with *AfupyrG* tail |
| OHS2264 | TCCCGGTAGATCTCATTCCC | 3' *AN10944* DR |
| OHS2265 | TCATCTGCCCAGGGATCG | 5' *AN10944* NF |
| OHS2266 | TGA GGG TAT CTG GGG TGG | 3' *AN10944* NR |
| OHS2267 | TCAGGGCCAAGATCACTGAG | 5' *AN10944* RT_F |
| OHS2268 | TACTGCTCCTGAGCTTCTCG | 3' *AN10944* RT_R |
| OHS2757 | CTCGGCCAGTTGTAGGTGT | 5' *rgsA* DF |
| OHS2758 | *GGCTTTGGCCTGTATCATGACTTCA*TCGTCCCCTTGACTGAACG | 3' *rgsA* with *AfupyrG* tail |
| OHS2759 | *TTTGGTGACGACAATACCTCCCGAC*TGGTATCACCACCACACCTT | 5' *rgsA* with *AfupyrG* tail |
| OHS2763 | *ACTTCTGCAGTCGGAATTGGCCTG*TCGTCCCCTTGACTGAACG | 3' *rgsA* with *AnipyroA* tail |
| OHS2764 | *TGGTGAGAACACATGCACAACTTG*TGGTATCACCACCACACCTT | 5' *rgsA* with *AnipyroA* tail |
| OHS2760 | ACCAGCTCGACATCCAACA | 3' *rgsA* DR |
| OHS2761 | GCACGAGTCCCTGTTCCT | 5' *rgsA* NF |
| OHS2762 | CTGCAGCGGTCCTCATCA | 3' *rgsA* NR |
| OHS2164 | CGACCTCGTCATCTTCGGAT | 5' *rgsA* RT_F |
| OHS2165 | CTGCGAGAGATAGGCCATGA | 3' *rgsA* RT_R |
| OHS1675 | AATT **GCGGCCGC** CTGCACCGCTACGACTAGG | 5' *mylA* with promoter and *Not*I_F |
| OHS1676 | AATT **GCGGCCGC** TGAGTCAGAGCTGGATTCGGA | 3' *mylA* with *Not*I_R |
| OHS0044 | GTAAGGATCTGTACGGCAAC | 5' *actin* RT_F |
| OHS0045 | AGATCCACATCTGTTGGAAG | 3' *actin* RT_R |
| OHS0580 | CAAGGCATGCATCAGTACCC | 5' *brlA* RT_F |
| OHS0581 | AGACATCGAACTCGGGACTC | 3' *brlA* RT_R |

## Table 2 (continued) | Oligonucleotides used in this study

| Name | Sequence (5'→3')ᵃ | Purpose |
|---|---|---|
| OHS0779 | ATTGACTGGGAAGCGAAGGA | 5' *abaA* RT_F |
| OHS0780 | CTGGGCAGTTGAACGATCTG | 3' *abaA* RT_R |
| OHS0803 | ACAGAGTTGCCTCAGCAGAT | 5' *wetA* RT_F |
| OHS0804 | AGATGTGCCTGTCTGGCTTA | 3' *wetA* RT_R |
| OHS0576 | GGTTGAAGTCGTCGGTTGAG | 5' *tpsA* RT_F |
| OHS0577 | TGGAAACCGATGAGGTCACA | 3' *tpsA* RT_R |
| OHS3014 | TCAGCGTCAAGGATGACAGT | 5' *tpsC* RT_F |
| OHS3015 | TCTTCACCTCGGTAGTCTGC | 3' *tpsC* RT_R |
| OHS1119 | GATTATTCGGCCCAGAGGGA | 5' *ccg9* RT_F |
| OHS1120 | ATGGCTTTCCACGTATTGGC | 3' *ccg9* RT_R |
| OHS3016 | CCCAATCAGGGAGCGATTTG | 5' *orlA* RT_F |
| OHS3017 | AGCCCTGATACTGCTTCTGG | 3' *orlA* RT_R |
| OHS3012 | TCTCAAGCCCTTCCCTCAAG | 5' *tppC* RT_F |
| OHS3013 | CCGCTTCTCAAACAAGCTGA | 3' *tppC* RT_R |
| OHS0578 | TGAGGAATTGACCACCGACA | 5' *fksA* RT_F |
| OHS0579 | GCACCAAGGATAGCAACAGG | 3' *fksA* RT_R |

ᵃTail sequences are shown in italics. Restriction enzyme sites are in bold.

### Quantitative reverse transcription-PCR (qRT-PCR)
To induce asexual development, the mycelia ball incubated for 18 hours was transferred to solid MM and cultured at 37 °C for designated time points[62]. The extracted total RNA was reverse-transcribed using GoScript Reverse transcriptase (Promega), and the resulting cDNA was used as a template for confirming the expression levels of target genes. The total qPCR system was determined with cDNA, target-specific primers, and iTaq Universal SYBR Green Supermix (BioRad, Hercules, CA, USA) on CFX96 Touch Real-Time PCR (BioRad). The target-specific primers are listed in Table 2. The expression levels were calculated using the $2^{-\Delta\Delta Ct}$ method, and the expression was normalized to that of β-actin, which was used as the control. All qRT-PCR experiments were conducted in triplicates.

### Spore stress analysis
Two-day-grown conidia were collected and diluted to a concentration of $10^3$ spores/mL. Each sample was heated at 55 °C for 30 min or incubated with 0.1 M $H_2O_2$ (Sigma, Burlington, MA, USA) for 30 min. All samples were diluted, spread onto solid MM, and incubated at 37 °C for 2 days. Relative survival rates were equal to the ratio of the number of viable colonies in the treated sample to the number of viable colonies in the untreated control. The spore stress response assay was repeated three times[62,63].

### Spore trehalose assay
Two-day-grown conidia were collected, diluted to $2 \times 10^8$ spores, and resuspended in 200 μL of distilled water. All samples were incubated at 95 °C for 20 min and centrifuged at 13,000 rpm for 10 min. Next, 100 μL of the supernatant was mixed with 100 μL of 0.2 M sodium citrate (pH 5.5, Sigma) and incubated at 37 °C for 8 h with or without trehalase (Sigma). The amount of trehalose, which was degraded by trehalase, was quantified as the amount of glucose using a Glucose Assay Kit (Sigma). The spore trehalose assay was repeated three times[55].

### Spore viability analysis
Conidia grown for 2, 7, or 10 days were collected and ten-fold diluted. Approximately 100 conidia were spread onto solid MM and incubated at 37 °C for 2 days. Relative survival rates were equal to the ratio of the number of viable colonies in the 10-day-grown conidia to the number of viable colonies in the 2-day-grown control. The spore viability assay was repeated three times[30].

**Article**

## Spore β-glucan analysis

To measure exposed glucan on the conidial surface, 2-day-grown conidia were collected and diluted to approximately $10^3$–$10^4$ conidia. Each sample was resuspended in 25 μL, mixed with 25 μL of Glucatell® reagent (Associates of Cape Cod, East Falmouth, MA, USA), and incubated at 37 °C for 30 min. Diazo reagent was added to all samples to stop the reaction, and optical density was measured at 540 nm. The spore β-glucan assay was repeated three times[64].

## Spore germination analysis

Approximately $1 \times 10^7$ 2-day-grown conidia were spread onto solid MM and incubated at 37 °C. Germination rates were measured every hour and calculated as the ratio of the number of germinated spores to the number of total spores in each sample. The spore germination assay was repeated three time[65].

## Statistics and reproducibility

All deletion mutants were generated independently in at least three strains. All experiments were repeated at least twice with three replicates each, and all data, except mutant construction, are reported as mean ± standard deviation. Student's unpaired $t$ test was used to evaluate statistical differences between the WT and Δ*mylA* strains. A $p$ value of <0.05 was considered statistically significant and >0.05 was considered not significant. All statistical analyses were conducted using the GraphPad Prism software (version 5).

## Reporting summary

Further information on research design is available in the Nature Portfolio Reporting Summary linked to this article.

## Data availability

All transcriptomic data are deposited in the National Center for Biotechnology Information database under the accession numbers PRJNA997009 and PRJNA927110. Source data underlying figures in the main text are available in Supplementary Data 3.

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

## Acknowledgements

The work by H.-S.P. was supported by the National Research Foundation of Korea (NRF) grant to H.-S.P. funded by the Korean government (MSIP: 2020R1C1C1004473). This research was supported by biological materials Specialized Graduate Program through the Korea Environmental Industry & Technology Institute (KEITI) funded by the Ministry of Environment (MOE) and supported by the Korea Basic Science Institute (National Research Facilities and Equipment Center) grant funded by the Ministry of Education (2021 R1A6C101A416).

## Author contributions

Y.-E.S. conceptualized, designed, performed, and analyzed the experiments. H.-J.C. participated in part experiments. Y.-E.S. and H.-S.P. wrote the manuscript, and H.-S.P. projected administration and supervision.

## Competing interests

The authors declare no competing interests.
