## [Peer Review File · Communications Biology]

Reviewers' comments:

Reviewer #1 (Remarks to the Author):

This manuscript describes the phenotypic analysis of the MyIA gene in *Aspergillus nidulans*. The following issues need to be addressed-

Abstract

Correct first sentence to -"myeloblastosis (MYB)-like proteins are a family of highly conserved transcription factors found in animals, plants..."

L27-29 are unclear. Did you study the family of Myb proteins, or just MyIA? Please rewrite clearly

Intro

The intro section L37-72 is too long, please shorten. The section L83-92 is too short.

L37 "The myeloblastosis (MYB)-like proteins include the..." change "include" to "contain a highly conserved..."

L74 "Lives ...even in air" I am not sure it lives suspended in the air, probably simply found in air samples as conidia etc.

L75 "higher efficiency in gene manipulation than other *Aspergillus* species" not sure this is correct nowadays with CRISPR.

L80 "possesses ambivalent characteristics and needs to be understood" unclear. What do you mean?

L95 "highly expressed in conidia compared with those in hyphae" what do you mean? What are "those" which genes/proteins?

Results

L104 "using InterPro" it is called InterProScan, please correct and provide ref.

L105-add they are shown in Table 1.

Fig. 2 please move to the supplementary, it is of secondary importance.

L113-126 are confusing and hard to understand. Please rewrite clearly.

L122 "the other seven proteins had two MYB repeats having R2 and R3 sequences," I cannot see this in Fig. 2, where is it?

L130- There are only seven 2R-mybs. To define their role in conidiogenesis, the best approach is to delete them and look at the phenotype. Trying to understand their involvement via gene expression is a much weaker approach. They may be important for conidiogenesis but unchanged in RNA level etc. This should be noted in the discussion.

L137- Please show the verification of the deletion and complementary strains in the supplement. Were several independent isolates verified for the deletion phenotype?

Fig 3D- please add the degree of conservation

3E- what are the time-points and growth conditions for V, A, C? please add.

L167- According to the methods, conidia were collected after 2 days of growth. Why 2 days, a very short time? Full maturation of conidia takes about 7 days.

Fig 5A, please add the number of genes found in each category.

L178- This is very general and probably indirect involvement of myIA in these processes. To really find what it does and narrow things down, you need to perform chip-seq analysis to see what genes it binds and possibly controls. Please add in discussion.

Fig 5F- you verified only 2 of the 5 genes shown in 5E, what was the verification result for the other 3 genes. Also please add a small map of the trehalose pathway showing the genes involved and the RNAseq changes in their expression in numerical fold change value.

Trehalose biosynthesis in *Aspergilli* is regulated by the transcription factors VosA and AtfA. How is their expression affected by myIA deletion?

Fig. 6- what about chitin synthesis genes? And mannosylation pathway genes?

6C- does your glucan assay measure exposed glucan on the conidial surface? Or do you break up the conidia and separate the cell wall before the assay? Please add.

7a- how many total number of genes were found in this cell cycle category? And how many are you showing here from that entire group? Please add.

Since mylA is strongly involved in conidiation, how does its deletion affect expression of other conidiation pathway genes (flbB, C, D, BrlA, VosA, AbaA, WetA etc). Where does it function within the known conidiation pathway?

Discussion

The discussion is too long. Please shorten it to the most important points and try not to repeat the results section.

L254 "CicD (transcriptional activator of cichorine)" correct to "CicD (transcriptional activator of cichorine biosynthesis)

L257 "the function of MYB-like proteins" correct to "the function of several MYB-like proteins"

L259 "we examined the total MYB-like proteins" you only looked at two and focused on one of them.

L271 "four conidia-specific" not conidia specific, they are enriched in conidia (vs. hypha)

L282-291- Move to results. Provide figure in supplement to show the various binding sites. Add if mylA deletion affects expression of other conidiation pathway genes (flbB, C, D, BrlA, VosA, AbaA, WetA etc).

L302-321, add the fold reduction of expression of each of the genes you describe here in your null mutant

L325-6 unclear, please rewrite

L333-346. Move to results.

L345- to prove this you will also need to do overexpression studies for these genes.

Reviewer #2 (Remarks to the Author):

In this manuscript the authors describe the identification and characterization of MylA, a MYB-like protein which appeared involved in fungal growth, conidiation, stress response and long-term viability of spores in *A. nidulans*. The study is very interesting and the experimental design is sound. However, I believe the language of the manuscript could use some improvement, in order to better convey the significance of the research.

Prior to publication there are several points that should be taken care of:

-the abstract and the introduction, more than the rest of the manuscript, would need a thorough English writing screen. (For example, it is unlikely that a protein family IS a transcription factor, as stated in line 24. In line 26 the MYB acronym is explained, but it first appeared in line 24. When talking about one specific protein identified, the correct article is THE and not a , again in line 26. And line 27, "of which" is referring to what? *A. nidulans*?) After careful reading and re-reading the sense come through, but the lack of correct English makes the text quite difficult to understand and follow. The abstract also does not explain clearly in which order things have been done, I firstly understood that initially the authors identified MylA and by characterizing this protein they connected it to the MYB-like family.

-in the introduction, various references are missing (line 39 and line 40, just to make an example. All the references about first discovery of the MYB domain, and the v-MYB causing leukemia in chickens. Or line 67).

-as non-expert in protein structure, the whole introductory part trying to explain the differences between 1R-2R and 3R MYB actually left me more confused. I acknowledge this might be due to my

lack of experience in this particular field, but if the paper should be accessible to a broad spectrum of scientist, I would suggest to simplify a bit this part.

-the paper occasionally suffers from short, repetitive sentences that could be combined for improved readability (especially in the results section).

-for the DEGs analysis, in the first experiment it's specified in the text (line 133) a p-value < 0.5, which seems quite high. Is this a typo, considering than later, in material and methods, the stated p-value is < 0.05? If not, why this high value? If yes, why write it again in the results if it is generally stated in the material and methods? Or the value stated in the material and methods section is valid only for the analysis mentioned in line 172?

-line 200, perhaps is worth mentioning more explicitly in the text that tpsA and ccg9 are threalose biosynthetic genes, otherwise might not be so clear why having those specific two genes there.

-the discussion part is very well written, and consider all aspects reported in the results. To be fair, it bings the argumentation even further, in fact there are parts that are not mentioned at all in the introduction or in the results part (like consideration about abaA deletion strain or LkhA). Maybe, if mentioning these part in the discussion, it would be better to report them as well in the results, and not let them, eventually, only in the supplementary material.

-in the material and methods, the statistical analysis for which particular experiments has been used? For the rest, this part in in general clear and understandable, with the exception of how the mutant strains have been generated.

Overall, this manuscript contains very good data and great science, but the words and style used to deliver the message are not appropriate. The discussion part is well written and thorough, but it seems out of place as the rest of the text is not at the same level. I would suggest to invest a bit more in the introduction and results part, to provide a more uniform style and better convey the nice discoveries achieved in this study.

Reviewer #3 (Remarks to the Author):

The paper by Ye-Eun Son et al. identified and characterized a myelobalstosis (MYB)-like protein MylA in the model organism *Aspergillus nidulans*. Further investigations showed MylA is essential for fungal growth and development, and regulating stress tolerance, cell wall integrity and long-term viability. These findings are data-detailed, credible, and innovative. It can be published in after addressing the following concerns.

1. Remove Fig.2 to supplemental files and concise the description for the conserved motifs.
2. Line 211-223, fksA was upregulated and β -glucan content higher in Δ mylA strain, indicating MylA inhibits the glucan biosynthesis. Why author thought MylA is required for β -glucan biosynthesis?

Reviewer #4 (Remarks to the Author):

Review on 'A MYB-like protein MylA contributes to conidiogenesis and conidial germination in *Aspergillus nidulans*' by Son YE et al

This manuscript focuses on MYB-like proteins of *A. nidulans* and specifically describes and characterizes the transcription factor MylA and its role in conidia and fungal growth. The manuscript is well organized and pleasant to read. Authors have used several molecular biology and biochemical techniques to identify and characterized the MylA of *A. nidulans*. They constructed Knock Outs and complementary strains, performed RNA seq, RT-qPCR, conidial trehalose analysis, as well as germination rates, phenotypic comparisons, conidial survival rates, among others. They have also used bioinformatics web servers to compare and describe the MYB-like proteins of *A. nidulans*.

I just have 2 comments that authors might address and others minor suggestions.

Comment 1. Lines 138, 139, 149.

In this study, deletion strains ($\Delta mylA$, $\Delta mylB$, $\Delta mylA/\Delta rgsA$) and complementary (C' mylA) strains have been generated. A supplementary figure is needed to show how these strains were constructed (schematic overview showing lengths and details of the assembled fragments used to generate the strains) and how they were confirmed (primers or restriction enzymes used for this).

Comment 2.

Authors should clarify some details of the RNA seq.

Line 133. Regarding the Fold Change (FC):

It is not clear if genes with fold change=2 were considered or not in the RNA seq analysis. It is written [fold change] >2 on line 133 but fold change ≥ 2 on line 454.

In line 454, authors are talking about DEGs (Differentially Expressed Genes) but is written 'fold change ≥ 2 ' which only consider UP-regulated genes.

In line 454 is written 'p value'. Please clarify whether the p-value or the adjusted p-value was used?

Minor suggestions:

Lines 333-336. I suggest to add a reference in this sentence (PMID: 15228532).

Line 336. It is important to add in which study the single mutant $\Delta rgsA$ was generated to know its parental background. The optimal case for Fig S3 is that the 4 strains compared have the same genetic background.

Line 324. The sentence starting with 'And continuously sequential cell cycle ...' needs revision. It is not clear.

Line 424. There is an error in the primer names given in 'using the primer pair OHS1675: OHS1675'. Two different primer names should be given.

Line 720. Fig. 1 does not have any subfigures, the label a is not required.

Reviewers' comments:

Reviewer #1 (Remarks to the Author):

This manuscript describes the phenotypic analysis of the MyIA gene in *Aspergillus nidulans*. The following issues need to be addressed-

→ We really appreciate valuable comments from this reviewer. We carefully checked the reviewer's comments and revised the manuscript accordingly. We hope that we have addressed all concerns raised by this reviewer.

Abstract

Correct first sentence to -“myeloblastosis (MYB)-like proteins are a family of highly conserved transcription factors found in animals, plants...”

→ Thank you for this comment. Following this comment, we revised the first sentence.

Lines 24-25: Myeloblastosis (MYB)-like proteins are a family of highly conserved transcription factors in animals, plants, and fungi, regulating the mRNA expression of genes.

L27-29 are unclear. Did you study the family of Myb proteins, or just MyIA? Please rewrite clearly.

→ Thank you for this valuable comment. Based on this comment, we revised this sentence.

Lines 25-27: In this study, we identified and characterized one of the MYB-like proteins in the model organism *Aspergillus nidulans*.

Introduction

The intro section L37-72 is too long, please shorten. The section L83-92 is too short.

→ Thank you for this comment. Following this comment, we intensively revised the introduction section.

L37 “The myeloblastosis (MYB)-like proteins include the...” change “include” to “contain a highly conserved.”

→ Thank you for this comment. Following this comment, we revised it.

Lines 58-60: The MYB-like proteins contain a highly conserved DNA-binding domain, the MYB domain, which consists of approximately 50 amino acids that are folded into three α -helices.

L74 “Lives ...even in air” I am not sure it lives suspended in the air, probably simply found in air samples as conidia etc.

→ Thank you for this comment. Following this comment, we revised it.

Lines 77-78: *A. nidulans* is a saprophytic filamentous fungus that lives ubiquitously in soil, crops and seeds, and even in water ²⁰.

L75 “higher efficiency in gene manipulation than other *Aspergillus* species” not sure this is correct nowadays with CRISPR.

→ Thank you for this comment. We removed this sentence while editing the introduction section.

L80 “possesses ambivalent characteristics and needs to be understood” unclear. What do you mean?

→ Thank you for this comment. We removed this sentence while editing the introduction section.

- Although *A. nidulans* possesses various characteristics and needs to be understood, the molecular mechanisms underlying fungal physiology remain largely unknown.

L95 “highly expressed in conidia compared with those in hyphae” what do you mean? What are “those” which genes/proteins?

→ Thank you for this comment. We carefully addressed this sentence.

Lines 91-92: characterized one *myb* gene that is highly expressed in conidia compared to hyphae.

Results

L104 “using InterPro” it is called interproscan, please correct and provide ref.

→ Thank you for this comment. Based on this comment, we revised it and added a reference.

Lines 99-100: To identify the MYB-like proteins in *A. nidulans*, we scanned the MYB-like domain using InterProScan tool ²⁵.

- 25 Jones, P. et al. InterProScan 5: genome-scale protein function classification. *Bioinformatics* 30, 1236-1240, doi:10.1093/bioinformatics/btu031 (2014).

L105-add they are shown in Table 1.

→ We added it.

Line 1011: in the *A. nidulans* genome (Table 1).

Fig. 2 please move to the supplementary, it is of secondary importance.

→ We appreciate this comment. Based on this comment, we moved Fig. 2 to Supplementary Figure 2.

Line 113: SWISS-MODEL (Supplementary Fig. 2).

L113-126 are confusing and hard to understand. Please rewrite clearly.

→ We really appreciate this comment. Following this comment, we revised these sentences.

Lines 110-117: The sequences of MYB motifs were aligned with those of *Arabidopsis thaliana*, investigated previously (Supplementary Fig. 1)¹⁶, and domain structures were predicted using SWISS-MODEL (Supplementary Fig. 2). As shown in Table 1, there were no MYB-like proteins having more than three MYB repeats. A total of 14 proteins had one MYB repeat, where 6 proteins had the R1/R2 motif, and 8 proteins had the R3 motif. In contrast, the other seven proteins had two MYB repeats having R2 and R3 motifs. The repeat size was 50 amino acids on average, and each repeat had three α -helices.

L122 “the other seven proteins had two MYB repeats having R2 and R3 sequences,” I cannot see this in Fig. 2, where is it?

→ Thank you for pointing it out. Based on this comment, we modified this figure.

L130- There are only seven 2R-mybs. To define their role in conidiogenesis, the best approach is to delete them and look at the phenotype. Trying to understand their involvement via gene expression is a much weaker approach. They may be important for conidiogenesis but unchanged in RNA level etc. This should be noted in the discussion.

→ We really appreciate this valuable comment. Following this comment, we added it in the discussion. We will also further characterize other MYB-like proteins in *A. nidulans*.

Lines 352-355: Additionally, although we focused on conidia-specific 2R MYB-like TFs, further studies are required to determine the function of other 2R or 1R MYB-like proteins in *A. nidulans* conidiogenesis.

L137- Please show the verification of the deletion and complementary strains in the supplement. Were several independent isolates verified for the deletion phenotype?

→ We thank for this critical comment. Following this comment, we added some figures of how mutant strains were generated and confirmed in supplementary Figures 4~6.

Son et al. Supplementary Fig. 4

Son et al. Supplementary Fig. 5

Son et al. Supplementary Fig. 6

Fig 3D- please add the degree of conservation

→ Thank you for this valuable comment. Based on this comment, we presented the degree of conservation in Fig. 2d and described in Figure legend.

Lines 728-737: **d** Phylogenetic tree and domain architecture of MyIA homologs in 5 *Aspergillus* species. A phylogenetic tree of MYB-like proteins existed in *A. nidulans* FGSC4, *A. fumigatus* Af293, *A. flavus* NRRL3357, *Aspergillus niger* and *Aspergillus oryzae* RIB40. All sequences were downloaded from FungiDB and aligned using Clustal W and MEGA X software, and a tree was generated in MEGA-X using the maximum likelihood method based on JTT matrix-based model. The bootstrap for this tree was repeated 1000 times. The numbers above the branches indicated the percentage of data coverage for internal nodes following the bootstrap test. The branch lengths, which were proportional to the numbers of nucleotide substitutions per site, were measured using a bar scale (0.2).

3E- what are the time-points and growth conditions for V, A, C? please add.

→ Thank you for this question. Based on this comment, we revised the figure and added growth conditions in Methods.

Lines 469-470: - To induce the asexual development the mycelia ball incubated for 18 hours were transferred to solid MM and cultured at 37 °C for designated time points

L167- According to the methods, conidia were collected after 2 days of growth. Why 2 days, a very short time? Full maturation of conidia takes about 7 days.

→ Thank you for this valuable comment. We always checked spore viability before we conduct phenotypic and transcriptomic analyses. As shown Figure 4g, the spore viability of 2-days grown $\Delta myIA$ conidia was nearly 100%, but the survival rate of 7- or 10-days grown $\Delta myIA$ conidia was decreased compared to conidia of WT or complemented strains. The viability of the $\Delta myIA$ conidia for 7 days old is only 60%. Therefore, when conducting experiments using this sample, it is difficult to accurately confirm the phenotype and gene expression. Therefore, we used conidia grown for 2 days for the experiments in this manuscript. We hope that this reviewer understands this situation.

Fig 5A, please add the number of genes found in each category.

→ We really appreciate this valuable comment. Following this comment, we added the number of genes in this figure and described in figure legend.

Lines 756-759: a The results of Gene Ontology analysis affected by MyIA. The upregulated (2320) or downregulated (1854) DEGs between WT and $\Delta myIA$ conidia were analyzed using the Gene Ontology term enrichment method. The gene numbers enriched in each category are indicated to the right of the points.

L178- This is very general and probably indirect involvement of *mylA* in these processes. To really find what it does and narrow things down, you need to perform chip-seq analysis to see what genes it binds and possibly controls. Please add in discussion.

→ Thank you for this valuable comment. We also agree with this reviewer's comment on direct roles of MylA in conidia. In current stage, we did not conduct ChIP-seq analysis, so we discussed it in the discussion section.

Lines 349-352: Despite our new findings, to provide novel insights into fungal development and spore properties, we will further investigate the direct regulatory mechanisms of MylA by performing Chromatin-Immuno-Precipitation (ChIP) sequencing and combining RNA-seq and ChIP-seq analyses.

Fig 5F- you verified only 2 of the 5 genes shown in 5E, what was the verification result for the other 3 genes. Also please add a small map of the trehalose pathway showing the genes involved and the RNA seq changes in their expression in numerical fold change value.

→ We appreciate this valuable comment. Following this comment, we verified mRNA levels of other 3 genes and we added a schematic illustration for trehalose synthetic pathway with RNA seq result.

Trehalose biosynthesis in *Aspergilli* is regulated by the transcription factors VosA and AtfA. How is their expression affected by *mylA* deletion?

→ Thank you for this valuable comment. We checked expression levels of *vosA* and *atfA* in WT and *mylA* mutant conidia. The log₂foldchange of *vosA* and *atfA* is about 0.36 and -2.23 in *mylA* deletion conidia, respectively. From this result, we thought MylA might affect mRNA level of *atfA*, but not *vosA*. The detailed mechanism will be further examined in the future.

Fig. 6- what about chitin synthesis genes? And mannosylation pathway genes?

→ This is a great question. We have already analyzed the expression of chitin synthetic genes but the mRNA levels of most of chitin synthetic genes were not significantly different between WT and $\Delta myIA$ conidia (data not shown). For the mannosylation pathway genes, we analyzed the galactomannan biosynthetic pathway genes and presented into Fig. 5d.

Lines 220-223: The mRNA expression levels of genes related to galactomannan, hydrophobins, and DHN-melanin biosynthesis were upregulated in *myIA* deletion conidia (Fig. 5d–5f). These results imply that MyIA can be involved in conidial wall composition.

6C- does your glucan assay measure exposed glucan on the conidial surface? Or do you break up the conidia and separate the cell wall before the assay? Please add.

→ Thank you for this comment. We did not break up the conidia, so the measured glucan might be exposed glucan on the conidial surface. We briefly mentioned it into the main text.

Lines 216-217: We then evaluated the β -glucan contents in the conidial surface of each strain.

7a- how many total number of genes were found in this cell cycle category? And how many are you showing here from that entire group? Please add.

→ Thank you for pointing it out. Following this comment, we added it.

Lines 229-233: To understand the effect of MyIA, we investigated the expression levels of 256 genes known to be associated with the mitotic cell cycle. As a result, a total of 95 genes were significantly differently expressed genes between WT and

$\Delta myIA$ conidia. Among them, about 80 % of them (76 DEGs/95 DEGs) were downregulated in the conidia of the $\Delta myIA$ strain (Fig. 6a).

Since *myIA* is strongly involved in conidiation, how does its deletion affect expression of other conidiation pathway genes (*flbB*, *C*, *D*, *BrlA*, *VosA*, *AbaA*, *WetA* etc). Where does it function within the known conidiation pathway?

→ This is a great comment. We agreed this valuable comment, so we checked mRNA levels of central development regulators (*BrlA*, *AbaA*, and *WetA*) after asexual induction. We presented this data into Figure 3d. In *myIA* deletion mutant conidia, we also checked mRNA levels of *flbB*, *flbC*, *flbD*, *vosA*, *brlA*, *abaA*, and *wetA*. mRNA levels of *flbC* ($\text{Log}_2\text{FC} = 2.08$), *flbD* ($\text{Log}_2\text{FC} = 2.35$), and *brlA* ($\text{Log}_2\text{FC} = 1.23$), were increased, but this is not necessary for in the main manuscript, so we did not mention it.

Lines 147-154: To investigate whether deletion of *myIA* affects expression of central development regulators, we examined the mRNA levels of *brlA*, *abaA*, and *wetA* after induction of asexual development in the $\Delta myIA$ mutant strain. As shown in Fig. 3d, *brlA* and *abaA* mRNA levels in $\Delta myIA$ strain were significantly increased at 24 h post asexual developmental induction compared to WT and C' *myIA* strains. And *myIA* deletion strain showed higher expression levels of *wetA* than WT and C' *myIA* strains at 48 h post induction. These results indicated that MyIA is needed for proper expression of asexual developmental regulators.

Discussion

The discussion is too long. Please shorten it to the most important points and try not to repeat the results section.

→ Thank you for this comment. Based on this comment, we intensively revised and shortened the discussion session. Also, several paragraphs were moved to the results section.

L254 “CicD (transcriptional activator of cichorine)” correct to “CicD (transcriptional activator of cichorine biosynthesis)

→ We revised it.

Line 270: and CicD (transcriptional activator of cichorine biosynthesis)^{12,38,39}.

L257 “the function of MYB-like proteins” correct to “the function of several MYB-like proteins”

→ We revised it.

Lines 266-267: the function of several MYB-like proteins in *Aspergillus* species remains unknown.

L259 “we examined the total MYB-like proteins” you only looked at two and focused on one of them.

→ We revised it.

Lines 267-268: We first analyzed the *A. nidulans* genomes and found 21 genes encoding MYB-like proteins (Fig. 1).

L271 “four conidia-specific” not conidia specific, they are enriched in conidia (vs. hypha)

→ We edited it.

Lines 272-274: In the present study, we selected two uncharacterized MYB-like proteins, MyIA and MyIB, which contain two MYB motifs (R2 and R3 repeats) based on RNA-seq data (Fig 2a).

L282-291- Move to results. Provide figure in supplement to show the various binding sites. Add if *myIA* deletion affects expression of other conidiation pathway genes (*flbB*, *C*, *D*, *BrlA*, *VosA*, *AbaA*, *WetA* etc).

→ Thank you for this great comment. Following this comment, we further investigated mRNA levels of the central developmental regulators (*BrlA*, *AbaA*, and *WetA*) in the *myIA* deletion mutants and described in result session. For the binding sites in the *myIA* promoter regions, we added it into the supplementary Figure 3.

Lines 147-154: To investigate whether deletion of *myIA* affects expression of central development regulators, we examined the mRNA levels of *brlA*, *abaA*, and *wetA* after induction of asexual development in the $\Delta myIA$ mutant strain. As shown in Fig. 3d, *brlA* and *abaA* mRNA levels in $\Delta myIA$ strain were significantly increased at 24 h post asexual developmental induction compared to WT and C' *myIA* strains. And *myIA* deletion strain showed higher expression levels of *wetA* than WT and C' *myIA* strains at 48 h post induction. These results indicated that MyIA is needed for proper expression of asexual developmental regulators.

Lines 279-296: In this study, we examined the role of MylA in asexual development. The $\Delta mylA$ strain exhibited decreased the amount of conidia and produced abnormal conidiophores (Fig. 3a). In addition, deletion of *mylA* affects mRNA expression of *brlA*, *abaA*, and *wetA* (Fig. 3d), implying that MylA might affect transcript levels of genes associated with asexual development. To confirm this hypothesis, in-depth mechanistic study will be needed. We found that mRNA level of *mylA* was specifically high in conidia (Fig. 2e), so we wondered which regulator controls mRNA level of *mylA* in late phase of asexual development or conidia. To test this, we first investigated the *mylA* promoter sequence and confirmed that there were one AbaA-response element (5'-CATTCTY-3'), one VelB-response element (5'-CCXTGG-3'), and two VosA-response elements (5'-CCXXGG-3'), but not WetA-response element (Supplementary Fig. 3a) 24,39. We next checked mRNA level of *mylA* in the deletion mutant strains. We checked the *mylA* transcripts levels with previous RNA-seq data and found no significant difference in levels in the $\Delta vosA$, $\Delta velB$, and $\Delta wetA$ strains 24. We then checked *mylA* transcript in the $\Delta abaA$ strain, which showed increased levels along the asexual development of the $\Delta abaA$ strain (Supplementary Fig. 3b). These suggest that AbaA, but not VosA, VelB, and WetA, might control *mylA* expression during asexual development of *A. nidulans*.

L302-321, add the fold reduction of expression of each of the genes you describe here in your null mutant

→ Thank you for this comment. Based on this comment, we described the fold change of each gene.

Lines 299-317: When we further analyzed the mitotic cell cycle-related genes, we found most genes to be downregulated in the conidia of the $\Delta mylA$ strain (Fig. 7a), among which LAMMER kinase A (**LkhA**; **Log2FC = -1.03**) affects the emergence of germ tube and hyphal polarity by modulating the transcript levels of *nimX^{cdc2}*, which controls the S phase and G2/M DNA damage checkpoint ⁴³. NimO (Never in mitosis; **Log2FC = -1.57**) is required to check a cell cycle linking S and M phases and is essential for proper DNA synthesis after progression of the S phase. *nimO* mutants exhibited much more delayed germination rate compared to that of WT strain 44. The putative NimO-binding partner Cdc7 (**Log2FC = -1.05**) also affects proper DNA replication ⁴⁵. As an anaphase-promoting complex/cyclosome (APC/C), BimA (Blocked-in-mitosis; **Log2FC = -1.81**) is essential for spindle poles to promote the onset of anaphase among mitosis. Loss-of-function mutations in *bimA* resulted in growth-arrested cells with condensed chromatin and a short mitotic spindle ⁴⁶. BimC,

a motor protein (kinesins; **Log2FC = -3.10**), plays a role in spindle pole body separation and mitotic spindle formation 47. SepA (**Log2FC = 2.07**) and SepH (**Log2FC = -1.13**) enable the formation of cortical actin rings at the surrounding septation and around the tips. In addition, PabA (**Log2FC = -1.29**) is required for proper septation during septation and cytokinesis in *A. nidulans* ⁴⁸. Altogether, MylA is required for activating almost all stages of mitosis, including the interphase and mitosis.

L325-6 unclear, please rewrite

→ We revised this paragraph.

Lines 319-325: Dormant conidia form germ tubes and synthesize septum after the first nuclear division. With our results, we hypothesize that MylA causes normal spore germination in *A. nidulans*. In fact, the absence of *mylA* led to a much less rate of conidial germination compared to that in the WT and C' *mylA* strains (Fig. 7c). These findings indicate that MylA plays a crucial role in conidial germination by regulating the cell cycle. Nonetheless, further studies are required to investigate the detailed molecular mechanisms of MylA in cell cycle.

L333-346. Move to results.

→ We really appreciate this valuable comment. Following this comment, we moved these sentences and revised the figure legend.

Lines 246-258:

Relationship between MylA and RgsA

It has been demonstrated that RgsA, a regulator of G-protein signaling, downregulates spore germination and vegetative growth by turning off the GanB-mediated signaling, which activates vegetative growth through the cAMP/PKA signaling pathway. We focused on the fact that the $\Delta rgsA$ mutant exhibited increased germination rates, unlike the $\Delta mylA$ mutant. To reveal the potential relationship between RgsA and MylA, we generated double deletion mutants of *mylA* and *rgsA* (Fig. 7a). As shown in Fig. 7b, the conidia of $\Delta mylA$ or $\Delta mylA\Delta rgsA$ strains appeared to exhibit poor spore germination ability. A quantitative analysis showed that the conidia of $\Delta mylA$ or $\Delta mylA\Delta rgsA$ strains exhibited considerably reduced germination rates, whereas the conidia of the $\Delta rgsA$ strain germinated rather quickly compared with the conidia of the WT strain (Fig. 7c). Remarkably, the germination rate of the double deletion mutant was similar to that of the $\Delta mylA$ mutant. These results suggested that MylA regulates conidial germination unaffected by deletion of *rgsA*, and the relationship between MylA and GanB-mediated signaling should be confirmed through additional studies.

L345- to prove this you will also need to do overexpression studies for these genes.

→ Thank you for this comment. We agreed this comment and revised this sentence.

Lines 256-258: These results suggested that MylA regulates conidial germination unaffected by deletion of *rgsA*, and the relationship between MylA and GanB-mediated signaling should be confirmed through additional studies.

Reviewer #2 (Remarks to the Author):

In this manuscript the authors describe the identification and characterization of MylA, a MYB-like protein which appeared involved in fungal growth, conidiation, stress response and long-term viability of spores in *A. nidulans*. The study is very interesting, and the experimental design is sound. However, I believe the language of the manuscript could use some improvement, in order to better convey the significance of the research.

Prior to publication there are several points that should be taken care of:

→ We really appreciate this reviewer's valuable comments for our manuscript. We carefully checked reviewer's comments and revised the manuscript following comments. We hope that we have improved the manuscript to a level of reviewer's satisfaction.

-the abstract and the introduction, more than the rest of the manuscript, would need a thorough English writing screen. (For example, it is unlikely that a protein family IS a transcription factor, as stated in line 24. In line 26 the MYB acronym is explained, but it first appeared in line 24. When talking about one specific protein identified, the correct article is THE and not a , again in line 26. And line 27, "of which" is referring to what? *A. nidulans*?) After careful reading and re-reading the sense come through, but the lack of correct English makes the text quite difficult to understand and follow. The abstract also does not explain clearly in which order things have been done, I firstly understood that initially the authors identified MylA and by characterizing this protein they connected it to the MYB-like family.

→ We really appreciate to comment. Following this comment, we carefully revised the abstract and the introduction sections. In addition, a professional editor carefully checked and revised the entire manuscript.

Lines 24-37: Myeloblastosis (MYB)-like proteins are a family of highly conserved transcription factors in animals, plants, and fungi, regulating the mRNA expression of genes. In this study, we identified and characterized one of the MYB-like proteins in the model organism *Aspergillus nidulans*. We screened mRNA levels of genes encoding the MYB-like proteins containing two MYB repeats in conidia, and found that mRNA levels of four genes including *flbD*, *cicD*, and two uncharacterized genes were high in conidia. To investigate the roles of two uncharacterized genes, *AN4618* and

AN10944, the deletion mutants for each gene were generated and found that *AN4618* was required for fungal development. Therefore, we further investigated the roles of *AN4618*, named as *mylA*, encoding the MYB-like protein containing two MYB repeats. Functional studies showed that MylA is essential for normal fungal growth and development. Phenotypic and transcriptomic analyses demonstrated that deletion of *mylA* affected stress tolerance, cell wall integrity, and long-term viability in *A. nidulans* conidia.

-in the introduction, various references are missing (line 39 and line 40, just to make an example. All the references about first discovery of the MYB domain, and the v-MYB causing leukemia in chickens. Or line 67).

→ We thank for this valuable comment. Following this, we added appropriate references in the introduction.

-as non-expert in protein structure, the whole introductory part trying to explain the differences between 1R-2R and 3R MYB actually left me more confused. I acknowledge this might be due to my lack of experience in this particular field, but if the paper should be accessible to a broad spectrum of scientist, I would suggest to simplify a bit this part.

→ We agreed this valuable comment, so we revised this part for the general reader.

Lines 58-69: The MYB-like proteins contain a highly conserved DNA-binding domain, the MYB domain, which consists of approximately 50 amino acids that are folded into three α -helices. Based on the number of imperfect repeats, MYB-like proteins are subdivided into four groups; 1R-MYB (one repeat), 2R-MYB (two repeats), 3R-MYB (three repeats), and 4R-MYB (four repeats). The 1R-MYB proteins are composed of a partial or single MYB repeat (R1/R2 or R3) and contribute to morphogenesis and development in plants^{13,14}. The 2R-MYB group is composed of two MYB motifs (R2 and R3 repeats), and most of these proteins are related to the determination of cell fate and regulation of cellular development, stress response, and primary and secondary metabolism in plants¹⁵. The 3R-MYB proteins have three MYB motifs, called R1, R2, and R3 repeats; this group regulates the cell cycle in humans, animals, and plants¹⁶⁻¹⁸.

-the paper occasionally suffers from short, repetitive sentences that could be combined for improved readability (especially in the results section).

→ We appreciate this comment. Following this comment, we revised the entire manuscript.

-for the DEGs analysis, in the first experiment it's specified in the text (line 133) a p-value < 0.5, which seems quite high. Is this a typo, considering that later, in material

and methods, the stated p-value is < 0.05? If not, why this high value? If yes, why write it again in the results if it is generally stated in the material and methods? Or the value stated in the material and methods section is valid only for the analysis mentioned in line 172?

→ We are so sorry for the confusion. We revised method for RNA-seq analysis.

Lines 448-450: DEGs were defined based on absolute value ($\log_2(\text{fold change}) \geq 1$ and p-value <0.05, and GO analyses were performed using the R package and FungiDB.

-line 200, perhaps is worth mentioning more explicitly in the text that *tpsA* and *ccg9* are trehalose biosynthetic genes, otherwise might not be so clear why having those specific two genes there.

→ Thank you for this valuable comment. We further checked mRNA levels of other genes involved in trehalose biosynthesis and added the results into the Figure 5f.

-the discussion part is very well written, and consider all aspects reported in the results. To be fair, it brings the argumentation even further, in fact there are parts that are not mentioned at all in the introduction or in the results part (like consideration about *abaA* deletion strain or *LkhA*). Maybe, if mentioning these part in the discussion, it would be better to report them as well in the results, and not let them, eventually, only in the supplementary material.

→ Thank you for this valuable comment. Following this comment, we moved some sentences to the results section. We also revised the discussion section to mention the relationship between *mylA* and other genes.

Lines 246-258: Relationship between MylA and RgsA/GanB signaling pathway in spore germination

It has been demonstrated that RgsA, a regulator of G-protein signaling, downregulates spore germination and vegetative growth by turning off the GanB-mediated signaling, which activates vegetative growth through the cAMP/PKA signaling pathway^{37,38}. We focused on the fact that the $\Delta rgsA$ mutant exhibited increased germination rates, unlike the $\Delta mylA$ mutant. To reveal the potential relationship between GanB/RgsA signaling and MylA, we generated double deletion mutants of *mylA* and *rgsA* (Fig. 7a). As shown in Fig. 7b, the conidia of $\Delta mylA$ or $\Delta mylA\Delta rgsA$ strains appeared to exhibit poor spore germination ability. A quantitative analysis showed that the conidia of $\Delta mylA$ or $\Delta mylA\Delta rgsA$ strains exhibited considerably reduced germination rates, whereas the conidia of the $\Delta rgsA$ strain germinated rather quickly compared with the conidia of the WT strain (Fig. 7c). Remarkably, the germination rate of the double deletion mutant was similar to that of the $\Delta mylA$ mutant. Collectively, we suggest that MylA acts downstream of GanB-mediated signaling for regulating conidial germination (Fig. 7d).

-in the material and methods, the statistical analysis for which particular experiments has been used? For the rest, this part is in general clear and understandable, with the exception of how the mutant strains have been generated.

→ Thank you for this valuable comment. Following this, we added the statistical analysis for the experiments in the methods section.

Lines 502-505: All data except mutant construction were reported as mean \pm standard deviation, and Student's unpaired *t*-test was used to evaluate statistical differences between the WT and $\Delta mylA$ strains. $p < 0.05$ was considered significant. All statistical analyses were conducted using the GraphPad Prism software (version 5)..

Overall, this manuscript contains very good data and great science, but the words and style used to deliver the message are not appropriate. The discussion part is well written and thorough, but it seems out of place as the rest of the text is not at the same level. I would suggest to invest a bit more in the introduction and results part, to provide a more uniform style and better convey the nice discoveries achieved in this study.

→ We truly appreciate this reviewer's positive and encouraging comments. We thoroughly revised the entire manuscript to understand the general readers. We hope that the revised version is suitable for publication in *Communication Biology*.

Reviewer #3 (Remarks to the Author):

The paper by Ye-Eun Son et al. identified and characterized a myelobalstosis (MYB)-like protein MylA in the model organism *Aspergillus nidulans*. Further investigations showed MylA is essential for fungal growth and development, and regulating stress tolerance, cell wall integrity and long-term viability. These findings are data-detailed, credible, and innovative. It can be published in after addressing the following concerns.

→ We really appreciate this reviewer's valuable comments for our manuscript. We carefully checked reviewer's comments and revised the manuscript following comments. We hope that we have improved the manuscript to a level of reviewer's satisfaction.

1. Remove Fig.2 to supplemental files and concise the description for the conserved motifs.

→ We truly appreciate this comment. Following this comment, we moved Figure 2 to Supplementary Figure 2.

2. Line 211-223, *fksA* was upregulated and β -glucan content higher in Δ *mylA* strain, indicating MylA inhibits the glucan biosynthesis. Why author thought MylA is required for β -glucan biosynthesis?

→ Thank you for this important question. Based on the RNA-seq data, we hypothesized that MylA regulates β -glucan synthesis, so we checked mRNA levels of genes associated with β -glucan synthesis and the amount of β -glucan in conidia. We found that MylA is required for proper β -glucan synthesis in *A. nidulans* conidia. We confirmed this conclusion through several experiments, but the detailed mechanism and why MylA regulates β -glucan biosynthesis are not clear. We also think that additional experiments are needed for the detailed mechanism. We briefly mentioned for this in the discussion section.

Lines 349-352: Despite our new findings, to provide novel insights into fungal development and spore properties, we will further investigate the direct regulatory mechanisms of MylA by performing Chromatin-Immuno-Precipitation (ChIP) sequencing and combining RNA-seq and ChIP-seq analyses.

Reviewer #4 (Remarks to the Author):

Review on 'A MYB-like protein MylA contributes to conidiogenesis and conidial germination in *Aspergillus nidulans*' by Son YE et al.

This manuscript focuses on MYB-like proteins of *A. nidulans* and specifically describes and characterizes the transcription factor MylA and its role in conidia and fungal growth.

The manuscript is well organized and pleasant to read. Authors have used several molecular biology and biochemical techniques to identify and characterized the MylA of *A. nidulans*. They constructed Knock Outs and complementary strains, performed RNA seq, RT-qPCR, conidial trehalose analysis, as well as germination rates, phenotypic comparisons, conidial survival rates, among others. They have also used bioinformatics web servers to compare and describe the MYB-like proteins of *A. nidulans*.

I just have 2 comments that authors might address and others minor suggestions.

→ We sincerely thank the positive comments from this reviewer. We carefully checked the reviewer's comments and revised the manuscript accordingly. We hope that we have addressed all issues raised by this reviewer.

Comment 1. Lines 138, 139, 149.

In this study, deletion strains ($\Delta mylA$, $\Delta mylB$, $\Delta mylA/\Delta rgsA$) and complementary (C' *mylA*) strains have been generated. A supplementary figure is needed to show how these strains were constructed (schematic overview showing lengths and details of the assembled fragments used to generate the strains) and how they were confirmed (primers or restriction enzymes used for this).

→ We really appreciate this valuable comment. Following this comment, we added the supplementary Figures S4-S6 on how to generate and confirm the mutant strains.

Son et al. Supplementary Fig. 4

Son et al. Supplementary Fig. 5

Comment 2.

Authors should clarify some details of the RNA seq.

Line 133. Regarding the Fold Change (FC):

It is not clear if genes with fold change=2 were considered or not in the RNA seq analysis. It is written |fold change| >2 on line 133 but fold change ≥ 2 on line 454.

In line 454, authors are talking about DEGs (Differentially Expressed Genes) but is written 'fold change ≥ 2 ' which only consider UP-regulated genes.

In line 454 is written 'p value'. Please clarify whether the p-value or the adjusted p-value was used?

→ We appreciate this comment. In response to this comment, we added details for the RNA-seq analysis, and revised the methods for the RNA-seq analysis.

Lines 448-450: DEGs were defined based on absolute value ($\log_2(\text{fold change}) \geq 1$ and p-value <0.05, and GO analyses were performed using the R package and FungiDB.

Minor suggestions:

Lines 333-336. I suggest to add a reference in this sentence (PMID: 15228532).

→ Thank you for pointing it out. Following this comment, we added a reference.

Lines 254-255: which activates vegetative growth through the cAMP/PKA signaling pathway^{37,38}.

Line 336. It is important to add in which study the single mutant $\Delta rgsA$ was generated to know its parental background. The optimal case for Fig S3 is that the 4 strains compared have the same genetic background.

→ We really appreciate this valuable comment. We generated the single mutants including $\Delta rgsA$ and $\Delta mylA$ using the recipient *A. nidulans* RJMP 1.59 (*pyrG89;pyrA4*) strain. For the $\Delta mylA \Delta rgsA$ strain, we used the $\Delta mylA$ (*pyrG89; $\Delta mylA::AfupyrG^+$; pyrA4*) strain for the recipient strain. In Figure S3, we used four strains, THS30 (*pyrG89; AfupyrG^+; pyrA4^+*), $\Delta mylA$ (*pyrG89; $\Delta mylA::AfupyrG^+$; pyrA4*), $\Delta rgsA$ (*pyrG89; $\Delta rgsA::AfupyrG^+$; pyrA4*), and $\Delta mylA \Delta rgsA$ (*pyrG89; $\Delta mylA::AfupyrG^+$; pyrA4; $\Delta rgsA::AnipyroA$*) strains which have the same genetic background. We described the generation of single or double deletion mutants in detail in Methods.

Lines 395-402: - For homologous fragments, the 5' and 3' regions of *mylA*, *mylB*, and *rgsA* were amplified using the primer set 5' DF: 3' *pyrG* tail and 5' *pyrG* tail: 3' DR, with *A. nidulans* FGSC4 genomic DNA (gDNA) as a template. The *pyrG* marker was amplified using the primer set OHS1542: OHS1543, with *A. fumigatus* Af293 gDNA as a template. All fragments were linked and amplified using the primer set 5' NF: 3' NR. The resulting PCR products were purified and introduced into the recipient RJMP 1.59 strain (*pyrG89; pyrA4*) according to a PEG-mediated transformation method

Lines 405-415: - The double deletion strains for $\Delta mylA \Delta rgsA$ were generated using the DJ-PCR method with the *pyroA* marker. The 5' and 3' regions of *mylA* were amplified using the primer set OHS2757: OHS2763 and OHS2764: OHS2760, respectively, and the *pyroA* marker was amplified using the primer set OHS1873: OHS1874, with *A. nidulans* FGSC4 gDNA as a template. All fragments were linked and amplified using the primer set OHS2761:OHS2762. The resulting PCR products were purified and introduced into the recipient $\Delta mylA$ strain TYE69.1 (*pyrG89; $\Delta mylA::AfupyrG$; pyrA4*) to produce TYE121.1~3. At least three independent double deletion strains were confirmed by PCR, followed by restriction enzyme digestion (Supplementary Figure 5). For analyzing phenotypes among single and double deletion strains, THS30 (*pyrG89; AfupyrG^+; pyrA4^+*) strain was used as the control.

Line 324. The sentence starting with 'And continuously sequential cell cycle ...' needs revision. It is not clear.

→ Thank you for this comment. We edited these sentences.

Lines 319-321: Dormant conidia form germ tubes and synthesize septum after the first nuclear division. With our results, we hypothesize that MylA causes normal spore germination in *A. nidulans*.

Line 424. There is an error in the primer names given in using the primer pair OHS1675: OHS1675'. Two different primer names should be given.

→ We are so sorry. It's a typo. So, we revised it.

Lines 417-419: For constructing the plasmid, the *mylA* gene region with its promoter was amplified using the primer pair OHS1675: OHS1676, digested with *NofI*, and cloned into pHS13⁶⁵.

Line 720. Fig. 1 does not have any subfigures, the label a is not required.

→ We appreciate this comment. Following this comment, we removed the label "a" in Figure 1.

REVIEWERS' COMMENTS:

Reviewer #1 (Remarks to the Author):

The authors have carefully revised the manuscript and answered all my comments. I still suggest careful editing of the English, especially new corrected sections. And then acceptance.

Reviewer #2 (Remarks to the Author):

In the revised version of this manuscript, the authors took great care in answering to every reviewers' comment. Major concerns have been addressed and resolved. The English language could still use little refining, but the text appeared now clearer and the meaning is anyway delivered.

Reviewer #3 (Remarks to the Author):

Dear all,

I really have appreciated the efforts made by the authors to answer to my comments and suggestions. However, I have a few comments in relation to response 1:

Supplementary Fig4:

b). myIA.

In the diagram on the top left. On the left part of the figure the cross line (in X) is not placed correctly. Please correct the diagram.

Supplementary Fig4:

b) myIB

I wonder if the agarose gels 1)DF-pyrG-R and 2)DF-RT-R are not switched, because the bands on the gels do not match the size of the bands described in the legend under these gels.

Example: For 1)DF-pyrG-R. Legend: wt: there should be no band but on the gel there is a 3.9 kb band
For 2)DF-RT-R Legend: wt: there should be a band of 3.9 kb band but on the gel there is no band.

Regarding the replied to my comment 2 and my suggestions, they have been well answered and I therefore accept these responses.

REVIEWERS' COMMENTS:

Reviewer #1 (Remarks to the Author):

The authors have carefully revised the manuscript and answered all my comments. I still suggest careful editing of the English, especially new corrected sections. And then acceptance.

→ We appreciate the reviewer accepting our response on the reviewer's comments. For the English language, we sent the revised manuscript to the specialized editor to address English language issues and hope that the revised manuscript is well-received by reviewers and general readers.

Reviewer #2 (Remarks to the Author):

In the revised version of this manuscript, the authors took great care in answering to every reviewers' comment. Major concerns have been addressed and resolved. The English language could still use little refining, but the text appeared now clearer and the meaning is anyway delivered.

→ We appreciate the reviewer accepting our response on the reviewer's comments. We acknowledge the English issues, and consequently, we sent the manuscript to the editor for refinement. We hope that the revised version is clearer and more readable for reviewers and general readers.

Reviewer #3 (Remarks to the Author):

Dear all,

I really have appreciated the efforts made by the authors to answer to my comments and suggestions. However, I have a few comments in relation to response 1:

→ We really appreciate this comment. The aspects pointed out by the reviewer were indeed our mistake, and we have revised all these issues. We sincerely appreciate your thorough review of our manuscript.

Supplementary Fig4:

b). mylA.

In the diagram on the top left. On the left part of the figure the cross line (in X) is not placed correctly. Please correct the diagram.

→ We corrected the diagram.

Supplementary Fig4:

b) mylB

I wonder if the agarose gels 1)DF-pyrG-R and 2)DF-RT-R are not switched, because the bands on the gels do not match the size of the bands described in the legend under these gels.

Example: For 1)DF-pyrG-R. Legend: wt: there should be no band but on the gel there is a 3.9 kb band

For 2)DF-RT-R Legend: wt: there should be a band of 3.9 kb band but on the gel there is no band.

→ We appreciate this comment. This is our mistake, so we revised this figure.